# EA-PS: Estimated Attack Effectiveness based Poisoning Defense in Federated Learning under Parameter Constraint Strategy

## Abstract

Federated learning is vulnerable to poisoning attacks due to the characteristics of its learning paradigm. There are a number of server-based and client-based poisoning defense methods to mitigate the impact of the attack. However, when facing persistent attacks with long-lasting attack effects, defense methods fail to guarantee robust and stable performance. In this paper, we propose a client-side defense method, EA-PS, which can be effectively combined with server-side methods to address the above issues. The key idea of EA-PS is to constrain the perturbation range of local parameters while minimizing the impact of attacks. To theoretically guarantee the performance and robustness of EA-PS, we prove that our methods have an efficiency guarantee with a lower upper bound, a robustness guarantee with a smaller certified radius, and a larger convergence upper bound. Experimental results show that, compared with other client-side defense methods combined with different server-side defense methods under both IID and non-IID data distributions, EA-PS reduces more performance degradation, achieves lower attack success rates and has more stable defense performance with smaller variance. Our code can be found at **https://anonymous.4open.science/r/EA-SP-6BC9**.

## 1 Introduction

Federated learning (FL) (Huang et al., 2023b) is a distributed machine learning paradigm that enables multiple parties to train models while preserving data privacy collaboratively. However, numerous studies (Lyu et al., 2023) have shown that malicious clients can manipulate the global model to result in significant damage.

Various defense strategies (Yin et al., 2018; Mhamdi et al., 2018; Blanchard et al., 2017) have been proposed to mitigate the impact of these attacks on the server-side, while fail to withstand strong attacks, such as persistent backdoor attacks (Liu et al., 2024) with long-lasting attack effects (Sun et al., 2021). To tackle the above issue, client-side defense methods provide more effective protection performance, combined with server-side defense methods. FL-WBC Sun et al. (2021) employs perturbations for defense, but their randomness can lead to a worse performance. To minimize the effect of attacks, LeadFL Zhu et al. (2023) is enhanced by utilizing hessian matrix. In this paper, we further enhance the objective function with a smaller optimization upper bound than LeadFL.

We empirically show that EA-PS⁻ (LeadFL with our objective function) has lower backdoor accuracy than FL-WBC and LeadFL with various server-side defense methods in IID and non-IID settings, as shown in Figure 1. More importantly, we observe that the backdoor defense performance of all three methods is unstable with large backdoor accuracy variances and distribution intervals. In addition, LeadFL and FL-WBC do not take into account the impact of untargeted attacks on the perturbation of model parameters.

Therefore, we propose a client-based defense approach named Estimated Attack Effectiveness based Poisoning Defense method under Parameter Constraint Strategy (EA-PS). It minimizes the long-lasting poisoning attack effect with a parameter constraint strategy to enhance stability by constraining the perturb range in the parameter space. We derive that our method has a smaller optimization upper bound and certified radius. Then, through Lagrangian relaxation and linear robust optimization, we integrate the constraints into the loss function to obtain an approximately optimal solution. Finally,

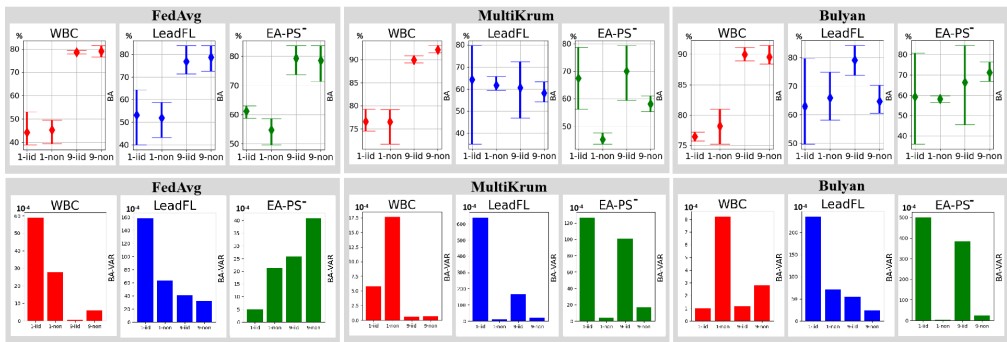

Figure 1: Defense performance of server-side defense methods under different attacks and data distributions on CIFAR10 dataset. The upper images are about the performance intervals of BA, and the other images are about the performance variances of BA.

by using a regularization method on parameter constraint, we increase the convergence upper bound while adaptively limiting the disturbance range of the parameter space. We evaluate our defense methods on CIFAR10 and FEMNIST (We also explore FashionMNIST in Appendix D) against the model poisoning attack under IID and non-IID settings. The results demonstrate that `EA-PS` can effectively mitigate the attack effect with stable defense performance.

Our key contributions are summarized as follows:

- We designed the `EA-PS` method, which effectively defends against poisoning attacks by minimizing the impact of long-lasting attacks and ensures the stability of the defense effect by the parameter constraint strategy.

- We derive a lower theoretical upper bound of the enhanced objective function to prove the efficiency of `EA-PS`. When implementing `EA-PS`, we further derive a robustness guarantee featuring a smaller certified radius and a larger convergence upper bound.

- We evaluate our defense methods on CIFAR10 and FEMNIST datasets under IID and non-IID settings against the model poisoning attacks with different server-side defense methods. Experimental results show that compared with other client defense methods, `EA-PS` can reduce more performance degradation by 0.79% for untargeted attacks, the backdoor success rate can be reduced by 14.9% at most for backdoor attacks with stability.

## 2 RELATED WORK

### 2.1 POISONING ATTACK IN FL

Model poisoning attacks can be classified into untargeted attacks (Lian et al., 2023b) and targeted attacks (Lyu et al., 2023). The objective of untargeted attacks is to disrupt the prediction accuracy of the model for any input, while targeted attacks aim to misclassify samples with specific triggers into categories chosen by the attacker. The attack mode we focus on in our work is poisoning attacks (targeted attacks in (Bagdasaryan & Shmatikov, 2020; Yang et al., 2025) and untargeted attacks in (Tolpegin et al., 2020; Fang et al., 2019)) with persistent attack strategies (Liu et al., 2024).

### 2.2 PRIOR ART ON DEFENSE METHODS

Server-side federated learning defenses fall into two categories: outlier detection/filtering (Huang et al., 2023a; Xu et al., 2025b) and robust aggregation (Huang et al., 2025). Filtering methods mitigate attacks by identifying and excluding malicious client model updates but may underutilize client information (Li et al., 2019a). Robust aggregation techniques are designed to identify and discard malicious updates, and mitigating their impact on the server.

Client-side defense methods provide more powerful protection performance combined with server-side defense methods. Existing client defense methods are divided into differential-privacy based methods (Naseri et al., 2020; Guo et al., 2024) and parameterized methods (Sun et al., 2021; Zhu et al.,

2023). When facing long-lasting attack effects, Parameterized methods perform better than differential-privacy based methods due to target and mechanism differences. Differential-privacy based methods not only have an uncontrollable defense effect but also unable to handle the accumulation of parameter pollution, and excessively large defense variance (Lian et al., 2023a; Bok et al., 2024; Bu & Liu, 2025). However, current parameterized approaches lack of tighter upper bounds and stable defense performance. Therefore, we propose the `EA-PS` method with lower upper optimization bounds, superior convergence properties and more stable defense performance compared to other methods.

## 3 MOTIVATION

To investigate the performance of current state-of-the-art methods (Sun et al., 2021; Zhu et al., 2023) with persistent attacks, we measured backdoor accuracy (BA) and their variance (VAR), as shown in Figure 1. Details about the results can be found in Appendix D. we can get two observations. 1) WBC Sun et al. (2021) method, designed with gradient constraint, has the most stable performance with the worst BA performance; 2) LeadFL (Zhu et al., 2023), designed with the constraint of gradient variation trend, has a better but not stable BA performance than WBC. Motivated by these, we design a new optimization method (Denoted as `EA-PS`⁻) with more historical information on the constraint of gradient variation trends, to minimize the impact of long-lasting attacks. The experimental results are shown in Figure 1 to verify that `EA-PS`⁻ is significantly improved compared with LeadFL, while still unstable. In addition, LeadFL, WBC and `EA-PS`⁻ do not take into account the impact of untargeted attacks on the perturbation of model parameters.

To ensure the stability of the defense effect and to improve the untargeted defense performance, we designed the parameter constraint strategy to ensure the stability in the optimization space. The simple idea of the parameter constraint strategy is to map the optimized manifold space of $A$ into the unit space $I$ by converting the spatial constraints into the base (rank) constraint $\lambda$ with spatial mapping $B$, reducing the dimensionality of the constraint space, and improving the efficiency of the constraint by simplifying the complexity of the parameter constraints.

## 4 MODEL POISONING ATTACK IN FL

### 4.1 PROBLEM FORMULATION

The aggregation objective of FedAvg is defined as follows:

$$\theta = \min_{\theta} \left\{ F(\theta) \triangleq \sum_{k=1}^{N} p^k F^k(\theta) \right\}, \tag{1}$$

where $\theta$ is the weights of the global model, $N$ represents the number of devices, $F^k$ is the local objective of the $k$-th device, $p^k$ represents the weight of the $k$-th device. In the $t$-th round of communication, the client updates the weights in the $e$-th round of local training as follows:

$$\theta_{t,e+1}^k \leftarrow \theta_{t,e}^k - \eta_{t,e} \bigtriangledown F(\theta_{t,e}^k), \tag{2}$$

where $\eta_{t,e}$ represents the learning rate and each local training round is updated on a mini-batch of data samples chosen from $k$-th client's data set. Finally, the server averages the parameters submitted by the $k$ models selected for aggregation (Zhu et al., 2023) as follows:

$$\theta^t \leftarrow \frac{N}{K} \sum_{k \in S_t} p^k \theta_t^k, \tag{3}$$

where $S_t$ is a set of participating clients in round $t$. $K$ is the number of selected clients by server-side defense methods. Based on FL-WBC Sun et al. (2021), define $\delta_t$ as the effect of the attack on the client in the $t$-th round as follows:

$$\delta_t = \frac{N}{K} \left[ \sum_{k \in \mathbb{S}_t} p^k \prod_{e=0}^{E-1} \left( \boldsymbol{I} - \eta_{t,e} \boldsymbol{H}_{t,e}^k \right) \right] \delta_{t-1}, \tag{4}$$

where $\boldsymbol{H}_{t,e}^k \triangleq \bigtriangledown^2 F(\theta_{t,e}^k) = (\theta_{t,e+1}^k - \theta_{t,e}^k - \Delta\theta_{t,e}^k)/\eta_t$ is the Hessian matrix at local iteration $e$ of global round $t$ and $I$ is the identify matrix.

For convenience, We define coefficient of attack impact $A_t$ as the relationship between two rounds as follows:

$$A_t \triangleq \sum_{k \in \mathbb{S}_t} p^k \prod_{e=0}^{E-1} \left( \boldsymbol{I} - \eta_{t,e} \boldsymbol{H}_{t,e}^k \right). \tag{5}$$

**Theorem 4.1.** *Minimizing $A_t - A_{t-1}$ yields a lower optimization upper bound than minimizing $A_{\hat{t}}$, where $A_{\hat{t}}$ is the coefficient of attack impact in LeadFL.*

First, the definition of $A_t = I - (P_t - P_{t-1}) + \Delta_t$, which is rearranged to derive $A_t - A_{t-1}$ with the assumption $A_t - A_{t-1} > \epsilon > 0$, and further used to obtain the recursive formula for $P_t = \sum_{i=1}^{t} \Delta_i + P_0 + [(t-1) + \varepsilon_{(P_{t-1})}]\varepsilon$.; similarly, the definition of $A_{\hat{t}} = I - (P_{\hat{t}} - P_{\hat{t}-1}) + \Delta_{\hat{t}}$ with $A_{\hat{t}} > \epsilon > 0$, which leads to the recursive formula for $P_{\hat{t}} = P_0 + \hat{t}I + \sum_{k=1}^{\hat{t}} \Delta_k + \hat{t}\varepsilon$. Then, substituting the recursive formula of $P_t$ into the definition of $A_t$ derives $A_t = I - (P_t - P_{t-1}) + \Delta_t = I + \Delta_1 - P_1 + P_0 - t\varepsilon$, simplifying $A_t$ to $I - (P_t - P_{t-1}) + \Delta_{\hat{t}}$, while substituting the recursive formula of $P_{\hat{t}}$ into the definition of $A_{\hat{t}}$ results in $I - (P_{\hat{t}} - P_{\hat{t}-1}) + \Delta_{\hat{t}} = I + \varepsilon$, reducing $A_{\hat{t}}$ to $I + \varepsilon$. Finally, calculating the difference $A_{\hat{t}} - A_t$ shows it equals $(t+1)\varepsilon > 0$, thus proving $A_t \leq A_{\hat{t}}$ and verifying Theorem 4.1. See Appendix B for a detailed proof.

## 4.2 PARAMETER CONSTRAINT STRATEGY

From the observations in Figure 1, it can be noticed that only minimizing the coefficient of attack impact ($A_t - A_{t-1}$ and $A_t$ ) can lead to unstable backdoor defense performance. Therefore, we propose a parameter constraint strategy that constraints $A$ to a parameter boundary (denoted as $\lambda$) to ensure that certain specific attacks are effectively detected while ensuring the stability of the parameters. In addition, the parameter constraint strategy can further alleviate the attack effect of untargeted attacks. The constraint equation is as follows:

$$\lambda I = B^{-1}AB \implies AB = \lambda B, \tag{6}$$

where $B$ is equivalent to $\delta$ as the spatial mapping for the effects of the attack.

## 5 EA-PS

### 5.1 DEFENSE DESIGN

The core idea of EA-PS is to achieve two goals:

- Goal 1: Minimize the impact of attacks to a better defense performance.
- Goal 2: Ensure the stability of defense performance by the parameter constraint strategy.

To achieve the first goal, we designed a new optimization objective function, namely $A_t - A_{t-1}$. To achieve the second goal, we designed the parameter constraint strategy to ensure the stability of the defense effect, namely $AB = \lambda B$.

$$\begin{aligned} Obj.min \quad & A_t - A_{t-1} \\ s.t. \quad & AB = \lambda B \\ & t > 1. \end{aligned} \tag{7}$$

Since $AB = \lambda B$ is the constraint, we assume that parameter boundary $\lambda$ is a linear set of $A$ based on the linear decision rule (Bertsimas et al., 2019). Without loss of generality, we define the following set:

$$\mathcal{L}^{T,N} = \left\{ \boldsymbol{A} \in \mathcal{R}^{T,N} \; \middle| \; \begin{array}{l} \exists \boldsymbol{A}_t, \boldsymbol{A}_{t-1}, \boldsymbol{t} \in [T_1] : \\ \lambda = \wp \boldsymbol{A}_t + \xi \boldsymbol{A}_{t-1} \end{array} \right\}, \tag{8}$$

where $\wp$ and $\xi$ are auxiliary variables. Then, the problem becomes equation (9) with an upper bound approximation to the near-optimal solution of the model (Ben-Tal et al., 2004).

$$\begin{aligned} Obj.min \quad & (A_t - A_{t-1}) + \alpha(AB - \lambda B) \\ s.t. \quad & \lambda \in \mathcal{L}^{T,N}, \\ & t > 1. \end{aligned} \tag{9}$$

Notably, the fixed $B$ in the above formulation fails to adapt to the varying spatial characteristics of $A_t$ and $A_{t-1}$. Then, we let $B$ adaptively change with coefficient $\beta$ to map better spatial space. And, we further denote $Regu$ as a regulation function to control the influence degree of $\beta$. The equation is as follows,

$$\begin{cases} B = \beta B, \\ Regu(\beta, B) = Max(\beta - \beta_{old}, 0.00001\beta), \end{cases} \tag{10}$$

where $\beta_{old}$ is $\beta$ of the previous round. And the $0.00001\beta$ term enforces a minimal effective update for $\beta$, ensuring that even small adjustments to $B$ are numerically distinguishable while preserving stability.

During computation, we employ an approximation by simplifying Equation (8) to $\lambda \simeq (A_t + A_{t-1})/2$. This approximation approach enables the equal fusion of the influence coefficients corresponding to the prior round ($t-1$) and the subsequent round ($t$). On one hand, it suppresses the cumulative effect induced by persistent attacks; on the other hand, the "stability objective" of the matching parameter constraint strategy can also preclude the deviation of the optimization space across rounds. We also showed its defensive stability in the ablation experiments (subsection 6.3). Then, the problem is approximated to Equation (11) as follows,

$$\begin{aligned} Obj.min \quad & (A_t - A_{t-1}) + \alpha(AB - \lambda B) + \gamma Regu(\beta, B) \\ s.t. \quad & \lambda \simeq (A_t + A_{t-1})/2 \\ & t > 1. \end{aligned} \tag{11}$$

To ensure the converge after above process, gradient trimming is performed during local training with a threshold $q$.

$$\text{clip}\left(\nabla\left(\mathbf{I} - \eta_{t,e}\widetilde{\mathbf{H}}_{t,e}^k\right), q\right)_{r,c} =$$

$$\begin{cases} \nabla\left(\mathbf{I} - \eta_{t,e}\widetilde{\mathbf{H}}_{t,e}^k\right)_{r,c}, & \left|\nabla\left(\mathbf{I} - \eta_{t,e}\widetilde{\mathbf{H}}_{t,e}^k\right)_{r,c}\right| \leq q, \\ q, & \left|\nabla\left(\mathbf{I} - \eta_{t,e}\widetilde{\mathbf{H}}_{t,e}^k\right)_{r,c}\right| > q, \end{cases}$$

where $r$ and $c$ are the indexes of rows and columns.

## 5.2 CONVERGENCE ANALYSIS

In this subsection, we derive convergence guarantees for FedAvg using `EA-PS` in the context of no malicious model attack. Specifically, for the $t$-th round, the local model on the $k$-th benign device is updated as:

$$\nabla F^{'}(\theta_{t,e}^k) = \nabla F(\theta_{t,e}^k) + \gamma Regu + clip. \tag{12}$$

Based on Assumptions B.1 to B.5 of the Appendix, we can derive the convergence guarantee of our defense on FedAvg(None) as follows.

**Theorem 5.1.** *(Convergence Guarantee): Let Assumptions B.1 to B.5 hold and $l, \mu, \sigma_k, G, K, N, \Gamma, F^*$ be defined therein and in Definition B.6. Choose $\kappa = \frac{l}{\mu}$, $\varphi = max(8\kappa, I)$ and the learning rate $\eta_t = \frac{2}{\mu(\varphi+t)}$. Then we have the following bound for `EA-PS`:*

$$E[F(\theta_T)] - F^* \leq \frac{\kappa}{\varphi + T - 1}\left(\frac{2(M+C)}{\mu} + \frac{\mu\varphi}{2}E[||\theta_0 - \theta^*||^2]\right), \tag{13}$$

where

$$D = E||\gamma Regu||_2^2, \tag{14}$$

$$C = \frac{N-K}{N-1}\frac{4}{K}E^2(d^2q^2 + G^2 + D^2), \tag{15}$$

$$M = \sum_{k=1}^N p_k^2(d^2q^2 + \sigma_k^2 + D^2) + 6l\Gamma + 8(I-1)^2(d^2q^2 + G^2 + D^2). \tag{16}$$

This proof begins by bounding the expected squared norm of the difference between the gradients before and after regularization, using the properties of the clip function and the definition of $D$ to show it is at most $(d^2q^2 + D^2)$. Leveraging this bound and Assumption B.3, along with the triangle

inequality, it derives that the variance of the modified gradient (i.e., the expected squared norm of the difference between the modified gradient and the true gradient) is bounded by $(d^2q^2 + \sigma_k^2 + D^2)$. Similarly, using Assumption B.4 and the same technique of splitting the norm difference, it obtains an upper bound of $(d^2q^2 + G^2 + D^2)$ for the expected squared norm of the modified gradient. Compared with LeadFL, `EA-PS` has a larger convergence upper bound, because of the parameter constraint and its regulation in Equation (12). **A higher tolerance to perturbations is achieved at the cost of a reduction in the convergence speed. It is noteworthy that in experiments, we set the same number of epochs as in LeadFL and still achieved better and more robust results.** See Appendix B for a detailed proof.

### 5.3 ROBUSTNESS ANALYSIS

In this subsection, we analyze the robustness of `EA-PS` using the certified radius framework proposed by (Panda et al., 2021) for the case of periodic attacks. We provide the definitions and assumptions in Appendix B.1. We consider a general threat model where the number of malicious clients in each round of attacks is random. Then, the certified radius of `EA-PS` combined with any given server-side defense is derived as:

**Certified Radius**: *Let $f$ be a c-coordinatewise-Lipschitz protocol on a dataset $\Omega$. Then $R(\rho) = \Lambda(T)(1 + dc)^{\Lambda(T)}\rho$ is a certified radius for $f$, where $\Lambda(t)$ is the cumulative learning rate $\Lambda(t) = \sum_{t=0}^{T-1} \eta_t$, $d$ is the dimension of model parameters.*

**Theorem 5.2.** *(Certified Radius): Let Assumption B.9 hold. The certified radius of the threat model is*

$$R(\rho) = (1 + dc)^{\sum_{t \in \Phi_T} \eta_t} \rho$$
$$(|\prod_{t \in \Gamma_T} \sum_{k \in S_t^*} p^k (\frac{N}{|S_t^*|} A_t)| + |\Phi_T| \sum_{t \in \Phi_T} \eta_t),$$

*where $\Phi_T$ is the set of communication rounds that server-side defenses cannot filter out all malicious updates. $\Gamma_T$ is the set of communication rounds that server-side defenses filter out all malicious updates. $S_t^*$ is a set of clients whose updates are not filtered out by the server-side defense in round $t$. $K_m^t$ is the number of malicious clients selected in round $t$. $g_{atk}$ is the probability that the server-side defense filters out all malicious updates versus the number of malicious clients selected in a communication round. $|\Phi_T|$ and $|S_t^*|$ are the cardinality of the set $\Phi_T$ and $S_t^*$, where $E[|\Phi_T|] = \sum_{t=0}^{T-1} g_{atk}(K_m^t)$.*

***Proof.*** Based on the definition of model updates, we use the triangle inequality to get the following inequality between $|\theta_t - \theta_t^*|$ and $|\theta_{t-1} - \theta_{t-1}^*|$ when the system is attacked in round $t - 1$.

$$|\theta_t - \theta_t^*| = |\theta_{t-1} - \eta_t \mu_t - \theta_{t-1}^* + \eta_t \hat{\mu}_t| \tag{17}$$

$$\leq |\theta_{t-1} - \theta_{t-1}^*| + \eta_t |\mu_t - \hat{\mu}_t|. \tag{18}$$

Using the triangle inequality again, we can get:

$$|\mu_t - \hat{\mu}_t| = |\mu_t - \mu_t^* + \mu_t^* - \hat{\mu}_t| \leq |\mu_t - \mu_t^*| + |\mu_t^* - \hat{\mu}_t|. \tag{19}$$

According to Definition B.7 and coordinate-wise Lipshitz in Assumption B.9:

$$|\mu_t - \hat{\mu}_t| \leq |\mu_t - \mu_t^*| + |\mu_t^* - \hat{\mu}_t| = dc|\theta_t - \theta_t^*| + \rho. \tag{20}$$

By plugging the above equation into Equation (18), we get:

$$|\theta_t - \theta_t^*| \leq |\theta_{t-1} - \theta_{t-1}^*| + \eta_t(dc|\theta_t - \theta_t^*| + \rho) \tag{21}$$

$$= (1 + dc\eta_t)|\theta_t - \theta_t^*| + \rho\eta_t. \tag{22}$$

According to Bernoulli's inequality, we have:

$$|\theta_t - \theta_t^*| \leq (1 + dc)^{\eta_t} |\theta_t - \theta_t^*| + \rho\eta_t. \tag{23}$$

Now we get the inequality between $|\theta_t - \theta_t^*|$ and $|\theta_{t-1} - \theta_{t-1}^*|$ when the system is attacked in round $t - 1$. Since we introduced server-side defense, we obtain the Equation (24) from the Equation (4):

$$\theta_t - \theta_t^* = \sum_{k \in S_t^*} p^k \left| \prod_{i=0}^{I-1} \left( \frac{N}{|S_t^*|} A_t \right) \right| (\theta_{t-1} - \theta_{t-1}^*). \tag{24}$$

Then we get the following relationship between $|\theta_t - \theta_t^*|$ and $|\theta_{t-1} - \theta_{t-1}^*|$ when server-side defense filters out all malicious updates in round $t-1$.

$$|\theta_t - \theta_t^*| \leq \sum_{k \in S_t^*} p^k \left| \prod_{i=0}^{I-1} \left( \frac{N}{|S_t^*|} A_t \right) \right| |\theta_{t-1} - \theta_{t-1}^*|. \qquad (25)$$

Finally, we can use Equations (23) and (25) to prove Theorem 5.2 by induction hypothesis:

$$R(\rho) = (1 + dc)^{\sum_{t \in \Phi_T} \eta_t} \rho(| \prod_{t \in \Gamma_T} \sum_{k \in S_t^*} p^k (\frac{N}{|S_t^*|} A_t)| + |\Phi_T| \sum_{t \in \Phi_T} \eta_t). \qquad (26)$$

Theorem 4.1 also indicates that `EA-PS` has a smaller coefficient of attack impact $A$ compared with LeadFL in the same environment. We have a smaller certified radius, as illustrated in Theorem 5.2.

## 6 EXPERIMENTS

We evaluate the client-side defense performances with multiple server-side defense methods on CIFAR10 dataset (Krizhevsky, 2009) under both IID and non-IID settings and on FEMNIST dataset (Caldas et al., 2018) under nature non-IID. We also explore the performance on FashionMNIST (Xiao et al., 2017) dataset (see Appendix D for detailed results). Our goal is to maintain main task accuracy and at the same time to achieve better and more stable backdoor defense performance. So, **main task accuracy(MA)**, **backdoor accuracy(BA)** and its **max**, **min**, **variance** are used in this paper. We use the 1/9-pixel attacks (Bagdasaryan & Shmatikov, 2020), spectrum attack (Wang et al., 2022), Gaussian attack (Tolpegin et al., 2020) and Label Flipping attack(Fang et al., 2019).

We use CMA & CTMA (Yin et al., 2018), Multi-Krum (Mhamdi et al., 2018), Bulyan (Blanchard et al., 2017) and alignins (Xu et al., 2025a) as server-side defense methods. For client-side defense methods, We choose FL-WBC Sun et al. (2021), LDP (Naseri et al., 2020), LeadFLZhu et al. (2023), LeadFL with our parameter constraint strategy (noted as `LeadFL+`) and `EA-PS-` as baseline methods. And the attack setting is the same as LeadFL Zhu et al. (2023) and FL-WBC Sun et al. (2021). The epoch is set to 80 for our experiments. In each round of training, 10 clients are randomly selected (1 to 5 of them are selected as malicious clients) to participate in the training. We set hyperparameters $\beta$ as 0.01, $\alpha$ as 0.1, $\lambda$ as 0.5, and $\gamma$ as 0.01. All baselines are built based on the source code of LeadFL Zhu et al. (2023). At the same time, our experimental results are obtained through five repeated experiments.

### 6.1 EFFECTIVENESS OF `LeadFL+`, `EA-PS-` AND `EA-PS`

Table 1 shows the defense results against on CIFAR10 and FEMNIST datasets. For the FEMNIST dataset, `LeadFL+`, `EA-PS-` and `EA-PS` methods outperform LeadFL in backdoor accuracy and its variance, which indicates that our methods can effectively defend against backdoor attacks. By comparing LeadFL with `LeadFL+` and `EA-PS-` with `EA-PS`, we observe that the parameter constraint strategy can improve the performance of BA by up to 22.55% and increase variance by up to 3.3% to ensure the effect and its stability. Comparing LeadFL with `EA-PS-` and `LeadFL+` with `EA-PS`, we find that the defense effect of `EA-PS-` and `EA-PS` is significantly higher than that of LeadFL and `LeadFL+`, which illustrate that the proposed new objective function can guarantee a more effective defense performance.

Meanwhile, we find that `LeadFL+`, `EA-PS-` and `EA-PS` methods have a more balanced performance on the CIFAR10 dataset compared with the FEMNIST dataset in the face of different server-side methods. By comparing LeadFL with `LeadFL+`, and `EA-PS-` with `EA-PS`, We find that the BA is improved by up to 14.9%, and the variance is improved by up to 40%. This also illustrates the effectiveness of our parameter constraint strategy. Comparing LeadFL with `EA-PS-` and `LeadFL+` with `EA-PS`, we get the same conclusion that the proposed new objective function can guarantee a more effective defense performance. Finally, comparing the improvement of `LeadFL+` and `EA-PS-` with respect to LeadFL individually, we get two observations. 1) `EA-PS-` has a significantly higher improvement in BA than `LeadFL+`, which indicates the proposed objective function has a better defense capability than the proposed parameter constraint strategy. 2) `EA-PS-` has a far less improvement of stability than `LeadFL+`, which indicates the proposed parameter constraint strategy has a significant capability to maintain defense stability. In addition, we conducted

comparative experiments on None (FedAvg), LDP (Naseri et al., 2020), and FL-WBC Sun et al. (2021) methods to prove the effectiveness of our proposed methods. The specific BA, variance, upper and lower bounds, and other information are shown in Table 10-16 of Appendix D.

| Data-set | distri-bution | server defense | client defense | Attack | | | | | | | | GAUS-SIAN | LabelFlip |
|---|---|---|---|---|---|---|---|---|---|---|---|---|---|
| | | | | 1-pixel | | 9-pixel | | Freq | | SADBA | | | |
| | | | | MA | BA | MA | BA | MA | BA | MA | BA | MA | MA |
| FEMNIST | NON-IID | FedAvg | LeadFL | 89.62 | 95.42/1.02 | 89.56 | 99.65/0.01 | 89.41 | 4.76/0.63 | 89.61 | 0.72/8.13 | 34.32 | 85.11 |
| | | | LeadFL+ | 89.24 | 95.23/0.63 | 89.37 | 99.59/0.01 | 88.53 | 4.82/0.01 | 89.12 | 0.56/0.17 | 33.81 | 84.1 |
| | | | EA-PS- | 89.45 | 92.95/0.69 | 89.41 | 94.7/1.42 | 88.69 | 4.63/0.21 | 89.00 | 0.11/7.24 | 33.62 | 84.12 |
| | | | EA-PS | 88.79 | 95.33/0.18 | 88.75 | 98.84/0.01 | 88.49 | 4.79/0.75 | 89.12 | 0.84/0.03 | 35.21 | 86.02 |
| | | CMA | LeadFL | 89.21 | 91.46/1.63 | 88.52 | 95.60/1.44 | 89.06 | 4.57/0.08 | 88.72 | 0.40/0.03 | 88.08 | 87.67 |
| | | | LeadFL+ | 88.16 | 89.56/1.32 | 88.86 | 96.34/1.52 | 88 | 4/0.07 | 89.14 | 1.66/1.56 | 88.19 | 86.91 |
| | | | EA-PS- | 88.28 | 88.92/1.40 | 88.30 | 86.66/1.12 | 87.97 | 4.44/0.15 | 88.81 | 1.90/0.20 | 87.09 | 86.93 |
| | | | EA-PS | 88.63 | 87.86/1.05 | 89.22 | 87.75/3.78 | 87.94 | 4.18/0.13 | 88.49 | 0.23/0.06 | 87.75 | 86.77 |
| | | multi-Krum | LeadFL | 88.31 | 65.75/21.54 | 88.38 | 59.01/49.62 | 88.57 | 4.61/0.83 | 88.65 | 2.12/2.36 | 86.12 | 87.7 |
| | | | LeadFL+ | 87.27 | 64.28/7.8 | 88.52 | 68.2/19.88 | 88.12 | 4.15/0.02 | 88.39 | 1.84/1.68 | 85.28 | 88.1 |
| | | | EA-PS- | 88.13 | 53.79/21.36 | 88.59 | 52.67/51.45 | 88.29 | 4.1/0.03 | 88.35 | 0.51/3.21 | 82.59 | 88.26 |
| | | | EA-PS | 87.76 | 43.2/7.68 | 88.48 | 53.84/16.33 | 87.92 | 4.24/0.17 | 89.12 | 0.21/0.01 | 85.82 | 87.95 |
| | | bulyan | LeadFL | 88.25 | 65.09/24.07 | 87.81 | 76.62/32.97 | 87.17 | 9.35/3.89 | 88.19 | 0.33/0.35 | 86.58 | 87.44 |
| | | | LeadFL+ | 87.76 | 57.95/10.38 | 86.40 | 71.68/22.9 | 87.03 | 7.62/2.69 | 87.62 | 0.37/0.02 | 86.55 | 87.23 |
| | | | EA-PS- | 87.46 | 50.92/28.90 | 87.58 | 74.93/17.2 | 87.52 | 4.21/0.04 | 88.09 | 0.04/2.63 | 86.27 | 87.66 |
| | | | EA-PS | 86.25 | 49.43/9.62 | 84.76 | 66.43/12.59 | 86.84 | 4.08/1.68 | 87.49 | 0.99/0.02 | 87.28 | 87.81 |
| | | alignins | LeadFL | 89.09 | 85.75/1.23 | 88.97 | 92.59/2.97 | 88.80 | 94.68/3.98 | 89.30 | 1.72/1.01 | 88.27 | 88.44 |
| | | | LeadFL+ | 88.60 | 83.25/1.66 | 88.84 | 76.33/3.25 | 88.87 | 85.45/5.03 | 88.75 | 2.71/0.05 | 87.72 | 88.21 |
| | | | EA-PS- | 88.54 | 83.30/1.36 | 88.68 | 74.60/60.01 | 87.76 | 87.76/20.65 | 88.75 | 1.3/0.06 | 87.93 | 88.46 |
| | | | EA-PS | 88.43 | 84.26/1.23 | 88.76 | 73.59/3.45 | 88.93 | 86.94/15.69 | 88.76 | 1.32/0.14 | 88.94 | 88.13 |
| CIFAR10 | IID | FedAvg | LeadFL | 40.67 | 53.09/158.59 | 36.41 | 76.9/41.04 | 34.05 | 56.73/16.28 | 33.34 | 4.26/1.87 | 33.93 | 16.01 |
| | | | LeadFL+ | 39.36 | 54.79/141.63 | 35.82 | 70.97/33.85 | 33.81 | 57.44/7.82 | 44.51 | 3.19/1.32 | 37.97 | 22.20 |
| | | | EA-PS- | 38.38 | 61.14/4.9 | 35.01 | 79.35/25.91 | 33.77 | 47.48/27.06 | 28.13 | 7.08/8.94 | 26.53 | 23.64 |
| | | | EA-PS | 39.79 | 54.52/34.95 | 44.17 | 77.8/6.69 | 34.76 | 46.55/3.83 | 33.03 | 4.49/0.67 | 38.87 | 21.93 |
| | NON-IID | | LeadFL | 37.64 | 51.84/63.03 | 45.08 | 78.78/32.38 | 30.18 | 85.02/84.48 | 29.02 | 8.94/25.62 | 19.08 | 16.16 |
| | | | LeadFL+ | 38.81 | 58.33/10.55 | 39.11 | 78/10.52 | 29.78 | 42.04/78.68 | 39.59 | 5.37/3.12 | 38.4 | 25.54 |
| | | | EA-PS- | 37.8 | 54.74/21.37 | 36.19 | 78.57/40.99 | 31.6 | 42.74/62.85 | 36.4 | 3.87/1.80 | 32.34 | 25.77 |
| | | | EA-PS | 40.31 | 50.6/15.6 | 43.22 | 77.1/5.52 | 31.04 | 47.57/77.29 | 35.69 | 6.15/1.24 | 32.43 | 24.58 |
| | IID | CMA | LeadFL | 40.14 | 46.54/301.31 | 36.45 | 78.34/14.66 | 31.15 | 45.78/13.68 | 34.93 | 6.60/5.29 | 28.79 | 15.15 |
| | | | LeadFL+ | 41.13 | 34.19/48.78 | 40.56 | 73.61/8.62 | 30.78 | 46.72/23.29 | 41.7 | 3.25/1.17 | 32.49 | 24.14 |
| | | | EA-PS- | 34.64 | 44.71/427.22 | 30.75 | 62.05/35.68 | 32.61 | 43.19/57.28 | 36.78 | 4.78/1.52 | 22.3 | 18.64 |
| | | | EA-PS | 39.8 | 32.58/0.02 | 42.9 | 63.57/0.01 | 35.84 | 42.43/2.14 | 36.87 | 11.51/1.60 | 33.87 | 16.92 |
| | NON-IID | | LeadFL | 37.39 | 31.47/2.35 | 32.79 | 61.21/11.06 | 28.01 | 33.83/45.05 | 31.77 | 6.51/5.17 | 17.19 | 18.71 |
| | | | LeadFL+ | 36.2 | 36.46/30.19 | 36.99 | 59.51/33.69 | 34.63 | 43.36/27.99 | 45.03 | 3.25/1.38 | 30.3 | 26.69 |
| | | | EA-PS- | 35.98 | 37.66/83.36 | 32.76 | 63.66/18.54 | 32.58 | 75.9/37.52 | 35.12 | 37.23/20.87 | 30.9 | 19.79 |
| | | | EA-PS | 41.36 | 33.18/91.01 | 44.47 | 63.21/3.35 | 31.79 | 41.02/6.87 | 43.16 | 3.88/2.82 | 34.50 | 18.57 |
| | IID | multi-Krum | LeadFL | 30 | 64.26/640.64 | 42.06 | 60.63/166.51 | 33.95 | 76.77/105.1 | 30.71 | 7.66/3.89 | 12.63 | 10.9 |
| | | | LeadFL+ | 32.88 | 61.72/62.41 | 40.29 | 63.39/5.2 | 32.64 | 77.31/42.75 | 39.16 | 10.21/0.85 | 15.17 | 19.36 |
| | | | EA-PS- | 31.05 | 67.4/126.97 | 32.12 | 32.12/100.78 | 33.79 | 74.66/14.63 | 35.49 | 60.27/0.01 | 13.05 | 20.64 |
| | | | EA-PS | 40.3 | 52.5/15 | 48 | 62.75/0.38 | 32.41 | 73.06/25.26 | 42.83 | 4.03/0.08 | 15.52 | 13.39 |
| | NON-IID | | LeadFL | 30.23 | 61.74/11.71 | 29.44 | 58.13/21.63 | 33.95 | 55.42/72.54 | 37.62 | 12.73/8.78 | 15.20 | 13.42 |
| | | | LeadFL+ | 33.4 | 56.84/26.28 | 30.22 | 61.55/14.22 | 37.72 | 42.12/36.09 | 41.7 | 4.92/1.44 | 23.15 | 13.35 |
| | | | EA-PS- | 36.63 | 45.2/4.54 | 38.12 | 53.42/16.76 | 35.6 | 45.92/69.23 | 33.13 | 7.08/13.95 | 12.49 | 13.39 |
| | | | EA-PS | 31.33 | 52.5/15 | 39.81 | 52.2/8.78 | 33.75 | 46.12/37.11 | 36.68 | 7.65/1.67 | 23.98 | 15.9 |
| | IID | bulyan | LeadFL | 34.75 | 63/236.4 | 37.66 | 79.16/54.75 | 33.44 | 40.85/52.83 | 39.35 | 5.51/0.71 | 13.01 | 14.27 |
| | | | LeadFL+ | 33.76 | 66.14/120.55 | 38.25 | 78.06/32.56 | 33.08 | 42.69/10.75 | 39.97 | 9.94/38.09 | 24.27 | 13.51 |
| | | | EA-PS- | 29.37 | 59.15/500.37 | 30.17 | 66.37/383.42 | 33.15 | 47.97/28.74 | 36.51 | 5.30/7.18 | 14.86 | 15.85 |
| | | | EA-PS | 33.88 | 59.11/7.61 | 30.25 | 61.3/0.04 | 32.18 | 49.76/12.17 | 37.25 | 9.25/7.71 | 36.38 | 14.8 |
| | NON-IID | | LeadFL | 32.31 | 65.96/71.8 | 37.3 | 64.65/25.71 | 32.1 | 41.33/7.62 | 37.79 | 12.77/46.70 | 20.69 | 18.94 |
| | | | LeadFL+ | 39.92 | 56.02/92.72 | 39.28 | 61.76/18.55 | 37.05 | 68.88/6.23 | 38.42 | 36.27/5.45 | 24.38 | 13.40 |
| | | | EA-PS- | 40.5 | 58.19/2.63 | 36.35 | 71.11/23.79 | 34.1 | 39.79/62.97 | 38.93 | 7.34/5.46 | 12.42 | 16.27 |
| | | | EA-PS | 43.12 | 59.11/7.61 | 38.55 | 66.05/15.58 | 34.81 | 40.13/7.81 | 36.64 | 8.83/3.73 | 27.62 | 16.08 |
| | IID | alignins | LeadFL | 37.25 | 71.79/0.04 | 32.87 | 62.02/13.39 | 39.18 | 62.23/17.20 | 32.88 | 3.40/1.98 | 12.41 | 17.88 |
| | | | LeadFL+ | 38.6 | 36.92/27.97 | 43.68 | 47.18/2.30 | 32.43 | 52.23/8.95 | 36.78 | 3.39/1.20 | 24.39 | 12.09 |
| | | | EA-PS- | 35.98 | 25.23/5.24 | 31.16 | 56.49/14.35 | 32.3 | 47.56/104.81 | 30.17 | 5.22/2.99 | 21.08 | 21.65 |
| | | | EA-PS | 29.46 | 27.51/7.83 | 25.71 | 55.57/5.33 | 37.29 | 60.15/13.56 | 41.07 | 4.85/0.34 | 32.08 | 26.36 |
| | NON-IID | | LeadFL | 32.99 | 32.37/8.39 | 37.28 | 60.15/13.56 | 36.83 | 48.35/122.83 | 34.34 | 10.92/125.66 | 30.58 | 12.94 |
| | | | LeadFL+ | 31.79 | 40.09/9.14 | 39.62 | 57.47/29.88 | 36.09 | 49.75/22.17 | 34.63 | 2.01/0.01 | 22.33 | 13.14 |
| | | | EA-PS- | 33.41 | 35.39/139.14 | 32.76 | 36.01/117.89 | 32.69 | 47.53/67.53 | 31.23 | 6.17/156.85 | 17.60 | 18.61 |
| | | | EA-PS | 22.19 | 32.09/0.16 | 39.75 | 57.79/12.96 | 42.05 | 54.15/15.44 | 34.5 | 3.43/2.02 | 31.75 | 17.07 |

Table 1: Comparison of main accuracy and backdoor accuracy on FEMNIST and CIFAR10 with IID/non-IID settings under poisoning attack. Where, the indicators are MA (%), BA (%/var($10^{-4}$)).

## 6.2 EFFECTIVENESS OF PARAMETER CONSTRAINT

The effect of parameter constraints depends on spatial mapping $\beta$ and regulation rate $\gamma$. Figure 2 shows EA-PS performance under 1/9-pixel attacks (IID and non-IID distributions) on CIFAR10: it is more stable under 9-pixel attacks than 1-pixel ones (due to weaker aggressiveness); BA performance is

more stable under IID than non-IID distributions (affected by data heterogeneity); attacks significantly impact defense stability; BA mean variance differences under different $\beta$ are $< 10\%$, indicating spatial mapping dynamic coefficients ensure defense stability. Additionally, $\gamma$ is more stable on non-IID, with optimal performance at $\gamma$=0.05. Experiments show that through the constraint of $\gamma$ on $\beta$, EA-PS can fully utilize the current round and historical attack data during training, effectively balancing the dynamic changes and cumulative impact of attacks. Tables 17 and 20 in Appendix D also show EA-PS is more stable across $\beta$ on FashionMNIST than CIFAR10.

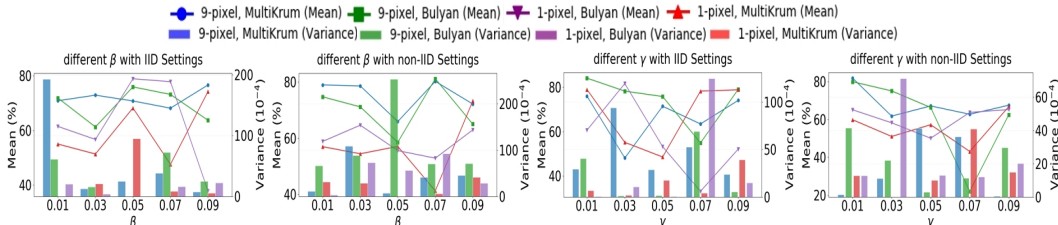

Figure 2: Impact of dynamic coefficient of the spatial mapping $\beta$ and regulation rate $\gamma$.

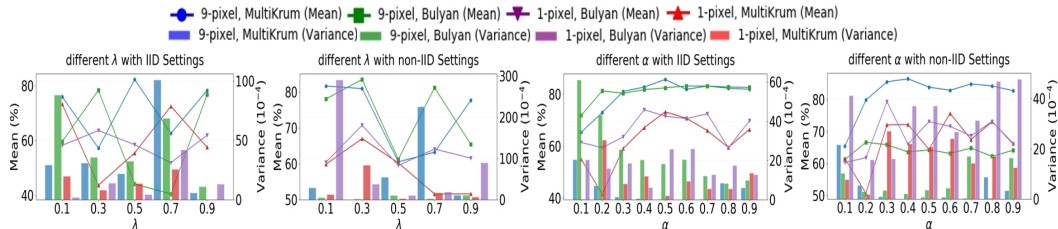

Figure 3: Impact of linear ratio in $\lambda$ and constraint rate $\alpha$.

## 6.3 EFFECTIVENESS OF THE SPATIAL MAPPING

The effect of parameter constraints is influenced by linear ratio $\lambda$ and constraint rate $\alpha$. Figure 3 shows the proposed method's performance on CIFAR10 under 1/9-pixel attacks (IID/non-IID distributions): 1) BA performance is more stable under 1-pixel than 9-pixel attacks; 2) No significant difference in BA stability between the two attacks; 3) Defense performance is most stable at $\lambda$=0.5, as it equally considers historical information of $A_t$ and $A_{t-1}$. For $\alpha$, the optimal range is 0.2-0.4. Larger $\alpha$ increases parameter constraint weight, biasing loss toward stability, but excessive weight raises variance via optimization space changes. This indicates the objective function ensures low BA, while parameter constraints guarantee effect stability. Consistent results are seen in FashionMNIST (Tables 18 and 19, Appendix D).

## 7 CONCLUSION

To defend against persistent attacks with long-lasting attack effects, we propose a client-side defense method, `EA-PS`, which can be effectively combined with server-side methods to guarantee robust and stable performance. Benefiting from minimizing the impact of attacks and the constraint of the perturbation range of local parameters, `EA-PS` method effectively thwarts backdoor poisoning attacks with stable performance. To theoretically guarantee the performance and robustness of `EA-PS`, we prove that our methods have a lower upper bound, a smaller certified radius, and a larger convergence upper bound. Evaluated on FEMNIST and CIFAR10 combined with different server-side defense methods under IID and non-IID data distributions, `EA-PS` reduces more performance degradation by 0.79%, achieves lower attack success rates by up to 14.9% and more stable defense performance with smaller variance by up to 40% compared with other client-side defense methods.

**Reproducibility Statement**

To ensure the reproducibility of the study, all key experimental information, theoretical derivation details, and technical resources are standardized and accessible, with specific references as follows:

Model and Algorithm Implementation: The core code of EA-PS is uploaded to an anonymous repository (https://anonymous.4open.science/r/EA-SP-6BC9), including complete parameter constraint strategies, gradient clipping mechanisms, and objective function optimization logic—consistent with Chapter 4 (Model Design) and Chapter 5 (Convergence and Robustness Analysis) of the main paper, enabling direct reproduction of experimental results.

Theoretical Derivation Verification: All assumptions (e.g., Assumptions B.1-B.9 in Appendix B), formula derivations, and key parameter definitions for EA-PS's optimization upper bound (Theorem 4.1), convergence guarantee (Theorem 5.1), and certified radius (Theorem 5.2) are detailed in the paper's appendix, facilitating verification of theoretical conclusions.

Experimental Design and Data: Datasets (CIFAR10, FEMNIST, FashionMNIST) with their sources, IID/non-IID division rules, attack sample construction (1/9-pixel, spectrum attacks), and hyperparameters ($\beta$=0.01, $\alpha$=0.1, $\lambda$=0.5, $\gamma$=0.01) are fully described in Chapter 6 (Experiments) and Appendix D. Detailed data (MA, BA, variance) for cross-validation is provided in Tables 10-20 of Appendix D.

Experimental Environment and Process: Training rounds (80), client selection rules (10 clients/round, 1-5 malicious), and server-side defense integration (CMA, Multi-Krum, Bulyan, alignins) are stated in Chapter 6. All results are averaged over 5 repeated experiments to ensure stability.

All materials above are available via the paper's main text, appendices, or the anonymous repository, supporting full reproduction of this study's results.

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

APPENDIX FOR

"EA-PS: ESTIMATED ATTACK EFFECTIVENESS BASED POISONING DEFENSE IN FEDERATED LEARNING UNDER PARAMETER CONSTRAINT STRATEGY"

In the appendix of this paper, we provide further details:

- In Appendix A, we show the detailed local training process after applying EA-PS and the comparison between server-side defense and EA-PS.

- In Appendix B, we make some definitions and assumptions for EA-PS. This includes mathematical assumptions, federated environment assumptions and definitions, and authentication radius definitions. Finally, we prove the three theorems we have presented.

- In Appendix C, we present the details of our experiments, including the dataset, the server-side & client-side defense methods, and the detailed experimental configuration, such as client selection and rounds, training details, model architectures, and evaluation metrics.

- In Appendix D, we show the detailed results (including main task accuracy, backdoor accuracy and their MAX, MIN and VAR values) of the baselines and proposed methods (including LeadFL$^+$, EA-PS$^-$, and EA-PS ) against 1/9-pixel, spectrum and label flipping attacks under IID and non-IID distributions on the FashionMNIST and CIFAR10 datasets.

- In Appendix E, we discuss the potential positive societal impacts of the work performed, the problem of **dataset selection**, **baseline selection**, **performance improvement**, **heterogeneity applicability** and **limitations** of the proposed method in this paper.

- In Appendix F, we state the use of EA-PS's large language models (LLMs).

## A. DETAILS OF ALGORITHM EA-PS

**Algorithm for training process applying EA-PS.** The detailed local training process of benign devices after applying EA-PS is shown in Algorithm 1.

**Comparison between server-side defense and EA-PS.** We analyze the differences in knowledge, capabilities, and assumptions between EA-PS and server-side defense, as shown in Table 2.

| Component | Client-Side Defense (Ours) | Server-Side Defense |
|---|---|---|
| Knowledge | Local model parameters and gradientsLocal training data distribution | Global aggregated modelAggregated update statistics (e.g., gradient norms) |
| Capability | Can apply local parameter masking/smoothing; Cannot modify server aggregation logic | Can modify aggregation rules (e.g., clip gradients, weight averaging) |
| Assumptions | Clients may be malicious Server is honest | Server is fully trustedClients may be malicious |

Table 2: Knowledge and capability comparison between server-side defense and EA-PS

## B. ASSUMPTIONS AND DEFINITIONS

**Assumption B.1** (Smoothness). $L$ is $\ell - smooth$ if $\forall x, y \in \Re^d$

$$L(x) - L(y) + (x - y)^T \bigtriangledown L(x) \leq \tfrac{\ell}{2} ||x - y||_2^2.$$

**Assumption B.2** (Convex). $L$ is $\mu - strongly$ convex if $\forall x, y \in \Re^d$

$$L(x) - L(y) + (x - y)^T \bigtriangledown L(y) \geq \tfrac{\mu}{2} ||x - y||_2^2.$$

**Assumption B.3** (Bound of Variance). Let $\xi_t^k$ be sampled from the $k$-th device's local data uniformly at random. The variance of the stochastic gradient in each device is bounded: $E|| \bigtriangledown L^k(\theta_t^k, \xi_t^k) - \bigtriangledown L^k(\theta_t^k)||^2 \leq \sigma_k^2$ for $k = 1, ..., N$.

**Assumption B.4** (Bound of Norm). The expected squared norm of stochastic gradients is uniformly bounded, i.e., $E|| \bigtriangledown L^k(\theta_{t,e}^k, \xi_{t,e}^k)|| \leq G^2$ for all $k = 1, ..., N$, $e = 0, ..., E - 1$ and $t = 0, ..., T - 1$.

---

**Algorithm 1** EA-PS and robust aggregation

---

**Input:** number of global rounds $T$, constraint rate $\alpha$, clipping bound $q$, $\sharp$ of clients selected in a round $K$, dynamic coefficient of the spatial mapping $\beta$, regulation rate $\gamma$.

**for** communication round $t = 0$ **to** $T - 1$ **do**

    Server randomly chooses $K$ clients;

    **parallel** $k = 0 \ldots K$ **do**

    Update model weights as global weights from the last round;

    **for** local iteration $e = 0, 1, \ldots$ **do**

        Compute gradients and update weights:

        $\theta_{t,e+1}^k \leftarrow \theta_{t,e}^k - \eta_{t,e} \bigtriangledown F(\theta_{t,e}^k)$;

        Compute the effect of poisoning attack:

        $A_t = \sum_{k \in \mathbb{S}_t} p^k \prod_{e=0}^{E-1} \left( \boldsymbol{I} - \eta_{t,e} \boldsymbol{H}_{t,e}^k \right)$;

        Compute the parameter boundary:

        $\lambda = (A_t + A_{t-1})/2$;

        Minimize the effect of poisoning attack and constraint the boundary of parameter:

        $A_t - A_{t-1} + \alpha(AB - \lambda B) + \gamma Regu(\beta, B)$;

        Compute and clip gradients:

        $\boldsymbol{R}_{t,e}^k = \mathrm{clip} \left( \nabla \left( \boldsymbol{I} - \eta_t \widetilde{\boldsymbol{H}}_{t,e}^k \right), q \right)$;

        Update weights;

        Update dynamic coefficient of the spatial mapping:

        $\beta_{old} \leftarrow \beta$;

    **end for**

    Compute updates;

    **end parallel**

    Aggregate updates using server-side defense;

    Update weights;

**end for**

---

**Assumption B.5** (Selection of Clients). Assume $S_t$ contains a subset of K indices uniformly sampled from [N] without replacement. Assume the data is balanced in the sense that $p_1 = \ldots = p_N = \frac{1}{N}$. The aggregation step of FedAvg performs $\theta_t \longleftarrow \frac{N}{K} \sum_{k \in S_t} p_k \theta_t^k$.

**Definition B.6** (Loss of clients). Denote $L^*$ and $L_k^*$ as the minimum value of $L$ and $L_k$, where $L$ is the loss of a model trained on the combination of datasets from all the clients and $L_k$ is the loss of a model trained on the dataset of client $k$. we can set $\Gamma = L^* - \sum_{k=1}^N p_k L_k^*$, which can quantify the degree of non-IID. If the data are IID, then $\Gamma$ goes to zero as the number of samples grows. If the data are non-IID, the $\Gamma$ is non-zero, and its magnitude reflects the heterogeneity of the data distribution.

**Definition B.7** (Poisoning Attack). For a protocol $f = (\mathcal{G}, \mathcal{A}, \eta)$ we define the set of poisoned protocols $F(\rho)$ to be all protocols $f^* = (\mathcal{G}^*, \mathcal{A}, \eta)$ that are exactly the same as $f$ except that the gradient oracle $\mathcal{G}^*$ is a $\rho - corrupted$ version of $\mathcal{G}$. That is, for any round $t$ and any model $\theta_t$ and any dataset $D$ we have $\mathcal{G}^*(\theta_t, D) = \mathcal{G}(\theta_t, D) + \epsilon$ for some $\epsilon$ with $||\epsilon||_1 \leq \rho$.

**Definition B.8** (Certified Radius). Let $f$ be a protocol and $f^* \in F(\rho)$ be a poisoned version of the same protocol. Let $\theta_T, \theta_T^*$ be the benign and poisoned final outputs of the above protocols. We call $R$ a certified radius for $f$ if $\forall f^* \in F(\rho); R(\rho) \geq |\theta_T - \theta_T^*|_1$.

**Assumption B.9** (Coordinate-wise Lipschitz). The protocol $f(\mathcal{G}, \mathcal{A}, \eta)$ is $c$-coordinate-wise Lipschitz if for any round $t \in [T]$, models $\theta_t, \theta_t^* \in \mathcal{M}$, and a dataset $D$ we have that the outputs of the gradient oracle on any coordinate can't drift too much farther apart. Specifically, for any coordinate index $i \in [d]$

$$|\mathcal{G}(\theta_t^*, D)[i] - \mathcal{G}(\theta_t, D)[i]| \leq c|\theta_t^* - \theta_t|_1.$$

B.10 PROOF OF THEOREM 4.1

**Theorem 4.1.** *Minimizing $A_t - A_{t-1}$ yields a smaller optimization upper bound than minimizing $A_{\hat{t}}$, where $A_{\hat{t}}$ is the coefficient of attack impact in LeadFL.*

*Proof.*

From the definition of $A_t$, we can get Equations (27) as follows:

$$A_t = I - (P_t - P_{t-1}) + \Delta_t. \tag{27}$$

According to Equation (27), we can obtain:

$$A_t - A_{t-1} = (\Delta_t - \Delta_{t-1}) - (P_t - 2P_{t-1} + P_{t-2}), \tag{28}$$

where we assume that the lower bound on the difference between $A_t - A_{t-1}$ and $0$ is $\epsilon$. For convenience, we consider this difference to be $\epsilon$.

According to Equation (28), we can obtain the recursion formula as follows:

$$\begin{cases} P_2 = \Delta_2 - \Delta_1 + 2P_1 - P_0 + \varepsilon \\ P_3 = \Delta_3 - \Delta_2 + 2P_2 - P_1 + \varepsilon \\ \quad\vdots \\ P_t = \Delta_t - \Delta_{t-1} + 2P_{t-1} - P_{t-2} + \varepsilon. \end{cases} \tag{29}$$

The recursive formula can be obtained from Equation (29) as follows:

$$\begin{cases} P_3 = \Delta_3 + \Delta_2 - 2\Delta_1 + 3P_1 - 2P_0 + 3\varepsilon \\ P_4 = \Delta_4 + \Delta_3 + \Delta_2 - 3\Delta_1 + 4P_1 - 3P_0 + 6\varepsilon \\ \quad\vdots \\ P_t = \sum_{i=1}^{t} \Delta_i + P_0 + [(t-1) + \varepsilon_{(P_{t-1})}]\varepsilon. \end{cases} \tag{30}$$

where $\varepsilon_{(P_{t-1})}$ represents the coefficient of the $\varepsilon$ in $P_{t-1}$. From the definition of $A_{\hat{t}}$, we can get Equations (31) as follows:

$$A_{\hat{t}} = I - (P_{\hat{t}} - P_{\hat{t}-1}) + \Delta_{\hat{t}}. \tag{31}$$

Similarly, assume that the lower bound on the difference between $A_{\hat{t}}$ and $0$ is also $\epsilon$.

According to Equation (31), we can obtain the recursion formula as follows:

$$\begin{cases} P_1 = P_0 + I + \Delta_1 + \varepsilon \\ P_2 = P_1 + I + \Delta_2 + \varepsilon \\ P_3 = P_2 + I + \Delta_3 + \varepsilon \\ \quad\vdots \\ P_{\hat{t}} = P_{\hat{t}-1} + I + \Delta_{\hat{t}} + \varepsilon. \end{cases} \tag{32}$$

The recursive formula can be obtained from Equation (32) as follows:

$$\begin{cases} P_2 = P_0 + 2I + \Delta_1 + \Delta_2 + 2\varepsilon \\ P_3 = P_0 + 3I + \Delta_1 + \Delta_2 + \Delta_3 + 2\varepsilon \\ \quad\vdots \\ P_{\hat{t}} = P_0 + \hat{t}I + \sum_{k=1}^{\hat{t}} \Delta_k + \hat{t}\varepsilon. \end{cases} \tag{33}$$

According to the Equation (28) and (30), we can obtain Equation (34) as follows:

$$A_t = I - (P_t - P_{t-1}) + \Delta_t = I + \Delta_1 - P_1 + P_0 - t\varepsilon. \tag{34}$$

According to the Equation (31) and (33), we can obtain Equation (35) as follows:

$$A_{\hat{t}} = I - (P_{\hat{t}} - P_{\hat{t}-1}) + \Delta_{\hat{t}} = I + \varepsilon. \tag{35}$$

We can obtain that $A_t$ has a lower upper bound than $A_{\hat{t}}$ as follows:

$$A_{\hat{t}} - A_t = (t+1)\varepsilon + P_1 - P_0 - \Delta_1 = (t+1)\varepsilon. \tag{36}$$

According to Equation (36), $A_t \leq A_{\hat{t}}$. Then we can get Theorem 4.1 that minimizing $A_t - A_{t-1}$ yields a smaller optimization upper bound than minimizing $A_{\hat{t}}$.

## B.11 PROOF OF CONVERGENCE GUARANTEE

**Theorem 5.1.** *(Convergence Guarantee): Let Assumptions B.1 to B.5 hold and $l, \mu, \sigma_k, G, K, N, \Gamma, F^*$ be defined therein and in Definition B.6. Choose $\kappa = \frac{l}{\mu}$, $\varphi = max(8\kappa, I)$ and the learning rate $\eta_t = \frac{2}{\mu(\varphi+t)}$. Then we have the following bound for* EA-PS:

$$E[F(\theta_T)] - F^* \le \frac{\kappa}{\varphi + T - 1}(\frac{2(M+C)}{\mu} + \frac{\mu\varphi}{2}E[||\theta_0 - \theta^*||^2]), \tag{37}$$

where

$$D = E||\gamma Regu||_2^2, \tag{38}$$

$$C = \frac{N-K}{N-1}\frac{4}{K}E^2(d^2q^2 + G^2 + D^2), \tag{39}$$

$$M = \sum_{k=1}^{N} p_k^2(d^2q^2 + \sigma_k^2 + D^2) + 6l\Gamma + 8(I-1)^2(d^2q^2 + G^2 + D^2). \tag{40}$$

*Proof.* The expected distance between the gradients before and after regularization can be bounded.

$$E|| \bigtriangledown L^{'}(\theta_{t,i}^k, \xi_{t,i}^k) - \bigtriangledown L(\theta_{t,i}^k, \xi_{t,i}^k)||_2^2 \tag{41}$$

$$= E||clip(A_t, q) + \gamma Regu||_2^2 \tag{42}$$

$$\le E||clip(A_t, q)||_2^2 + E||\gamma Regu||_2^2 \tag{43}$$

$$\le d^2q^2 + D^2. \tag{44}$$

Using the bounds above and Assumption B.3, we can derive new bounds for the variance of modified gradient $E|| \bigtriangledown L^{'}(\theta_{t,i}^k, \xi_{t,i}^k) - \bigtriangledown L(\theta_{t,i}^k)||^2$

$$E|| \bigtriangledown L^{'}(\theta_{t,i}^k, \xi_{t,i}^k) - \bigtriangledown L(\theta_{t,i}^k)||^2 \tag{45}$$

$$\le E|| \bigtriangledown L^{'}(\theta_{t,i}^k, \xi_{t,i}^k) - \bigtriangledown L(\theta_{t,i}^k, \xi_{t,i}^k)||^2 \tag{46}$$

$$+ E|| \bigtriangledown L(\theta_{t,i}^k, \xi_{t,i}^k) - \bigtriangledown L(\theta_{t,i}^k)||^2 \tag{47}$$

$$\le d^2q^2 + \sigma_k^2 + D^2, \tag{48}$$

where we use the triangle inequality. Similarly, we can also derive bounds for the expected squared norm of modified gradients using Assumption B.4.

$$E|| \bigtriangledown L^{'}(\theta_{t,i}^k, \xi_{t,i}^k)||^2 \tag{49}$$

$$\le E|| \bigtriangledown L^{'}(\theta_{t,i}^k, \xi_{t,i}^k) - \bigtriangledown L(\theta_{t,i}^k, \xi_{t,i}^k)||_2^2 \tag{50}$$

$$+ E|| \bigtriangledown L(\theta_{t,i}^k, \xi_{t,i}^k)||^2 \tag{51}$$

$$\le d^2q^2 + G^2 + D^2. \tag{52}$$

Applying the bounds for the variance and the expected squared norm of modified gradients after applying EA-PS, we can derive our convergence guarantee from Theorem 5.1 in Li et al. (2019b) by replacing these bounds. Compared with LeadFL, the EA-PS method has a larger convergence upper bound, because of the parameter constraint and its regulation in Equation (12).

## B.12 PROOF OF CERTIFIED RADIUS OF THE THREAT MODEL

Our paper uses the same Poisoning Attack Definition (Definition B.7) and Coordinate-wise Lipschitz Assumption (Assumption B.9) as LeadFL.

**Theorem 5.2. Certified Radius of the Threat Model:** EA-PS *has a smaller upper bound and its certified radius is also smaller than LeadFL.*

*Proof.*

From the proof of the certified radius in LeadFL, it is known that under the assumption of unification, `EA-PS` has a smaller upper bound and its certified radius is also smaller than LeadFL. Based on the definition of model updates, we use the triangle inequality to get the following inequality between $|\theta_t - \theta_t^*|$ and $|\theta_{t-1} - \theta_{t-1}^*|$ when the system is attacked in round $t-1$.

$$|\theta_t - \theta_t^*| = |\theta_{t-1} - \eta_t\mu_t - \theta_{t-1}^* + \eta_t\hat{\mu}_t| \leq |\theta_{t-1} - \theta_{t-1}^*| + \eta_t|\mu_t - \hat{\mu}_t|. \tag{53}$$

Using the triangle inequality again, we can get:

$$|\mu_t - \hat{\mu}_t| = |\mu_t - \mu_t^* + \mu_t^* - \hat{\mu}_t| \leq |\mu_t - \mu_t^*| + |\mu_t^* - \hat{\mu}_t|. \tag{54}$$

According to Definition B.7 and coordinate-wise Lipshitz in Assumption B.9:

$$|\mu_t - \hat{\mu}_t| \leq |\mu_t - \mu_t^*| + |\mu_t^* - \hat{\mu}_t| = dc|\theta_t - \theta_t^*| + \rho. \tag{55}$$

By plugging the above equation into Equation (53), we get:

$$|\theta_t - \theta_t^*| \leq |\theta_{t-1} - \theta_{t-1}^*| + \eta_t(dc|\theta_t - \theta_t^*| + \rho) = (1 + dc\eta_t)|\theta_t - \theta_t^*| + \rho\eta_t. \tag{56}$$

According to Bernoulli's inequality, we have:

$$|\theta_t - \theta_t^*| \leq (1 + dc)^{\eta_t}|\theta_t - \theta_t^*| + \rho\eta_t. \tag{57}$$

Now we get the inequality between $|\theta_t - \theta_t^*|$ and $|\theta_{t-1} - \theta_{t-1}^*|$ when the system is attacked in round $t-1$. Since we introduced server-side defense, we obtain the Equation (58) from the Equation (4):

$$\theta_t - \theta_t^* = \sum_{k \in S_t^*} p^k \left| \prod_{i=0}^{I-1} \left( \frac{N}{|S_t^*|} A_t \right) \right| (\theta_{t-1} - \theta_{t-1}^*). \tag{58}$$

Then we get the following relationship between $|\theta_t - \theta_t^*|$ and $|\theta_{t-1} - \theta_{t-1}^*|$ when server-side defense filters out all malicious updates in round $t-1$.

$$|\theta_t - \theta_t^*| \leq \sum_{k \in S_t^*} p^k \left| \prod_{i=0}^{I-1} \left( \frac{N}{|S_t^*|} A_t \right) \right| |\theta_{t-1} - \theta_{t-1}^*|. \tag{59}$$

Finally, we can use Equations (57) and (59) to prove Theorem 5.2 by induction hypothesis:

$$R(\rho) = (1 + dc)^{\sum_{t \in \Phi_T} \eta_t} \rho(| \prod_{t \in \Gamma_T} \sum_{k \in S_t^*} p^k(\frac{N}{|S_t^*|} A_t)| + |\Phi_T| \sum_{t \in \Phi_T} \eta_t). \tag{60}$$

From Theorem 4.1., we have a smaller coefficient of attack impact $A$ compared with LeadFL in the same environment. So, we have a smaller certified radius.

## C. EXPERIMENTS DETAIL

### C.1. DATASETS

We conduct experiments on FashionMNIST, CIFAR10 and FEMNIST (nature NON-IID). In the case of FashionMNIST, every one of the 100 clients is allocated 600 images from a total of 60,000 images. As for CIFAR10, each client obtains 500 images out of the 50,000 available images. As for FEMNIST, each client obtains 433 images.

In the IID setting, samples are uniformly distributed to clients. In the non-IID setting, we deploy the limited label strategy (McMahan et al., 2016) that is also used for the evaluation of LeadFL in FashionMNIST and CIFAR10: Of the 10 classes in each of the two datasets, each client is assigned 5 random classes. They are then assigned an equal number of randomly selected samples from each of their classes. The clients' datasets are selected independently.

For the regularization term, we tune the parameters of the Dirichlet distribution in the non-IID case using hyper-parameters $\alpha$. Here we set $\alpha = 0.4$ in FashionMNIST and FEMNIST and 0.25 in CIFAR10.

## C.2. SERVER-SIDE & CLIENT-SIDE DEFENSES

We use CMA&CTMA (Yin et al., 2018), Multi-Krum (Mhamdi et al., 2018), Bulyan (Blanchard et al., 2017) and alignins(Xu et al., 2025a) as server-side defenses.

For client-side defenses, we choose FL-WBC (Sun et al., 2021), LDP (Naseri et al., 2020), LeadFL, LeadFL with our parameter constraint strategy (noted as `LeadFL`$^+$) and `EA-PS`$^-$ as the baseline. For FL-WBC (Sun et al., 2021) and LDP (Naseri et al., 2020) defenses, we apply Laplace noise with $mean = 0$ and $std = 0.2$ as in the original papers. For LeadFL and `EA-PS`$^-$, we set the clipping norm $q = 0.2$. For `LeadFL`$^+$, we set the dynamic coefficient of the spatial mapping $\beta = 0.05$, regularization rate $\alpha = 0.1$ and linear ratio of $\lambda$ to 0.5.

## C.3. CONFIGURATIONS

**Client Selection and Rounds.** There are 100 clients in total, of which 25 are malicious. There are 80 global rounds and 10 local rounds. The server selects 10 clients per global round. For most experiments, the selection is random but consistent over experiments, i.e., for two experiments, the clients selected in round t are the same to enable comparison between the different settings.

**Training Details.** For training, we set local epoch $E$ as 1 and batch size $BS$ as 32. We apply SGD optimizer and set the learning rate $\eta$ to 0.01. Up to five clients are selected as malicious clients in each round of 80 communication rounds. Our hyper-parameters are **(1)** $\beta$ is the initial value of adaptive parameter constraint, set to 0.01. **(2)** $\alpha$ is the ratio of adaptive parameter constraint to `EA-PS`$^-$, set to 0.1. **(3)** $\lambda$ is the ratio of linear decision rules, set to 0.5. **(4)** $\gamma$ is the ratio of regulation to control the influence degree of $\beta$, set to 0.01.

**Model Architectures.** We adopt the same model architecture as LeadFL (Zhu et al., 2023) on the FashionMNIST and CIFAR10 with convolutional layers and fully-connected layers. Meanwhile, a similar network structure is adopted for the FashionMNIST dataset on the FEMNIST dataset.

**Evaluation Metrics.** Our goal is to maintain main task accuracy and at the same time achieve better and more stable backdoor defense performance. So, **Main Task Accuracy(MA)**, **Backdoor Accuracy(BA)** and its **MAX**, **MIN**, **Variance** are used in this paper. (1)**Main Task Accuracy(MA)**: We measure the main task accuracy using the accuracy of the global model on the benign test set of the main task. As in other works, we consider the maximum accuracy achieved during training. (2)**Backdoor Accuracy(BA)**: Backdoor accuracy measures how successful an attacker is in integrating the backdoor into the model. We measure the accuracy of the backdoor as the percentage of samples with triggers that are classified as attacker intent. We find that the backdoor accuracy does not converge in our experiments, so we consider the average backdoor accuracy. And because the difference in backdoor accuracy is large in each round, we use the average backdoor accuracy in our experiments. At the same time, we also give the **MAX** and **MIN** of the experimental results, which represent the defense effect interval of the `EA-PS` method. (3)**Backdoor Accuracy Variance**: We measure the backdoor accuracy variance to represent the stability of the defense effect.

## C.4. COMPUTE WORKER

The computing device used in our experiments is an RTX3090 GPU and a CPU with 60G memory.

# D. ADDITIONAL RESULTS

Table 3 shows the results of different client-side defense methods combined with different server-side defense methods in the **FashionMNIST** dataset under **IID** distribution in the case of **9-pixel** attacks.

Table 4 shows the results of different client-side defense methods combined with different server-side defense methods in the **FashionMNIST** dataset under **non-IID** distribution in the case of **9-pixel** attacks.

Table 5 shows the results of different client-side defense methods combined with different server-side defense methods in the **FashionMNIST** dataset under **IID** distribution in the case of **1-pixel** attacks.

Table 6 shows the results of different client-side defense methods combined with different server-side defense methods in the **FashionMNIST** dataset under **non-IID** distribution in the case of **1-pixel** attacks.

Table 7 shows the results of different client-side defense methods combined with different server-side defense methods in the **FashionMNIST** dataset under **IID** distribution in the case of **spectrum** attacks.

Table 8 shows the results of different client-side defense methods combined with different server-side defense methods in the **FashionMNIST** dataset under **non-IID** distribution in the case of **spectrum** attacks.

Table 9 shows the results of different client-side defense methods combined with different server-side defense methods in the **FashionMNIST** dataset under **IID** and **NON-IID** distribution in the case of **Label Flipping** attacks.

Table 10 shows the results of different client-side defense methods combined with different server-side defense methods in the **CIFAR10** dataset under **IID** distribution in the case of **9-pixel** attacks.

Table 11 shows the results of different client-side defense methods combined with different server-side defense methods in the **CIFAR10** dataset under **non-IID** distribution in the case of **9-pixel** attacks.

Table 12 shows the results of different client-side defense methods combined with different server-side defense methods in the **CIFAR10** dataset under **IID** distribution in the case of **1-pixel** attacks.

Table 13 shows the results of different client-side defense methods combined with different server-side defense methods in the **CIFAR10** dataset under **non-IID** distribution in the case of **1-pixel** attacks.

Table 14 shows the results of different client-side defense methods combined with different server-side defense methods in the **CIFAR10** dataset under **IID** distribution in the case of **spectrum** attacks.

Table 15 shows the results of different client-side defense methods combined with different server-side defense methods in the **CIFAR10** dataset under **non-IID** distribution in the case of **spectrum** attacks.

Table 16 shows the results of different client-side defense methods combined with different server-side defense methods in the **CIFAR10** dataset under **IID** and **non-IID** distribution in the case of **Label Flipping** attacks.

Table 17 shows the experimental results of tuning the hyper-parameter $\beta$ on the **FashionMNIST** dataset under 1/9-pixel backdoor attacks.

Table 18 shows the experimental results of tuning the hyper-parameter $\alpha$ on the **FashionMNIST** dataset under 1/9-pixel backdoor attacks.

Table 19 shows the experimental results of tuning the hyper-parameter $\lambda$ on the **FashionMNIST** dataset under 1/9-pixel backdoor attacks.

Table 20 shows the experimental results of tuning $\gamma$ on **FashionMNIST** with IID/non-IID settings under 1/9-pixel backdoor attacks.

Table 21 shows the experimental results of time **overhead**.

## E. DISCUSSIONS AND LIMITATIONS

**Potential positive societal impacts of the work performed:** Our research focuses on the defense of federated learning poisoning attacks, which ensures data privacy and system stability. It not only protects public safety and business fairness, but also clears the way for the implementation of federated learning technology.

**Dataset Selection:** Due to the limitation of the paper's length, we used two commonly used datasets for verification. Although our experimental results perform better on the FashionMNIST dataset, the CIFAR10 dataset is superior to the FashionMNIST dataset in terms of complexity, generalization, and scene diversity, we mainly adopt the experimental results of the CIFAR10 dataset in the analysis of the main text.

| Server-side Defense | Client-side Defense | MA | BA | VAR |
|---|---|---|---|---|
| None | None | 89.88 | 98.53 (+0.36 / -0.35) | 0.13 |
| | LDP | 88.36 | 90.81(+1.33 / -1.06) | 1.47 |
| | WBC | 88.23 | 90.30(+0.86 / -1.26) | 1.24 |
| | LeadFL | 87.42 | 93.22(+1.5/ -3.48) | 7.75 |
| | LeadFL$^+$ | 87.39 | 89.36(+1.73/-2.26) | 3.66 |
| | EA-PS$^-$ | 87.11 | 88.8(+2.35/ -1.22) | 1.86 |
| | EA-PS | 86.92 | 90.72(+1.8 / -2.13) | 3.02 |
| CMA | None | 89.79 | 96.65(+0.49 /-0.83) | 0.52 |
| | LDP | 87.03 | 96.66(+1.59 /-1.30) | 2.15 |
| | WBC | 87.19 | 96.78(+0.72 / -0.37) | 0.39 |
| | LeadFL | 87.72 | 95.2(+1.64/ -2.64) | 2.8 |
| | LeadFL$^+$ | 87.22 | 93.62(+1.55/-1.43) | 2.71 |
| | EA-PS$^-$ | 86.87 | 92.73(+3.3/ -2.39) | 6.91 |
| | EA-PS | 86.98 | 91.93(+2.36/ -1.97) | 4.23 |
| CTMA | None | 89.86 | 96.32(+0.38 / -0.61) | 0.29 |
| | LDP | 88.26 | 98.18(+0.11 / -0.06) | 0.01 |
| | WBC | 88.2 | 97.8(+0.18 /-0.17) | 0.03 |
| | LeadFL | 87.32 | 87.42(+4.55/ -6.17) | 15.04 |
| | LeadFL$^+$ | 86.82 | 83.65(+4.36/-5.62) | 16.77 |
| | EA-PS$^-$ | 86.67 | 79.38(+4.85/ -7.36) | 29.5 |
| | EA-PS | 86.76 | 84.65(+4.11/ -4.63) | 13.31 |
| multiKrum | None | 89.40 | 33.59(+37.15 /-18.95) | 1034.74 |
| | LDP | 86.78 | 76.41(+2.17 /-1.89) | 4.18 |
| | WBC | 86.83 | 77.52(+0.52 / -0.52) | 0.27 |
| | LeadFL | 86.72 | 32.82(+7.56 / -5.52) | 23.4 |
| | LeadFL$^+$ | 86.38 | 23.34(+5.15/-4.33) | 19.7 |
| | EA-PS$^-$ | 86.08 | 21.73(+6.7/ -9.75) | 52.13 |
| | EA-PS | 86.18 | 22.97(+3.29/ -3.16) | 7.95 |
| bulyan | None | 89.4 | 36.23(+46.58 /-23.93) | 1627.7 |
| | LDP | 85.88 | 74.34(+3.42/-2.74) | 9.84 |
| | WBC | 85.96 | 73.93(+4.16 /-6.82) | 35.29 |
| | LeadFL | 85.73 | 32.78(+10.54/ -13.99) | 99.34 |
| | LeadFL$^+$ | 85.44 | 22.57(+10.81/-9.39) | 84.13 |
| | EA-PS$^-$ | 85.03 | 19.7(+25.72/ -13.56) | 246.97 |
| | EA-PS | 85.77 | 19.59(+6.67/ -11.42) | 65.19 |

Table 3: Comparison under 9-pixel pattern backdoor attack on IID FashionMNIST dataset

**Performance Improvement:** As can be seen from Table 17-20 in the appendix and the default experimental settings of this paper, there still exists much room for defense performance improvement of the proposed method through fine-tuning of hyperparameters. However, current experiments and theoretical analyses are sufficient to prove the superiority of the proposed method. Additionally, although our method has a larger convergence upper bound, we set the same number of epochs in the experiment for a fair comparison with methods such as LeadFL and WBC. To further improve defense performance, our method can appropriately increase the number of epochs.

**Baseline Selection:** In this paper, the proposed method is compared with the latest parameterized client-side methods. There are new developments in client-side methods from 2024 to 2025, but most of them are based on differential privacy, distillation learning, or malicious client detection, which means they are not comparable to the proposed method.

**Heterogeneity Applicability:** Our method is based on the client backdoor defense method, so our method can also perform backdoor defense under heterogeneity, but it needs to be adjusted adaptively according to the structure of different client networks.

**Limitations:** We present an experimental analysis of the time overhead to illustrate the main limitation of EA-PS. The experimental results, in Table 21, show that EA-PS combined with various server defense methods is higher than LeadFL on CIFAR10 and FEMNIST datasets. This is because adaptive

| Server-side Defense | Client-side Defense | MA | BA | VAR |
|---|---|---|---|---|
| None | None | 89.69 | 97.97(+0.87 / -0.56) | 0.59 |
| | LDP | 87.88 | 88.24(+1.42 / -0.93) | 1.59 |
| | WBC | 88.05 | 89.4(+1.56 / -1.18) | 1.9 |
| | LeadFL | 87.21 | 92.43(+1.47 / -2.06) | 2.29 |
| | LeadFL$^+$ | 86.24 | 89.04(+1.56/-1.69) | 2.24 |
| | EA-PS$^-$ | 86.43 | 88.10(+3.54 / -8.27) | 13.98 |
| | EA-PS | 86.48 | 86.67(+2.77/ -0.676) | 2.15 |
| CMA | None | 89.67 | 95.07(+0.32 /-0.51) | 0.2 |
| | LDP | 86.86 | 95.18(+0.31 /-0.25) | 0.2 |
| | WBC | 86.3 | 96.1(+1.98 / -1.06) | 2.94 |
| | LeadFL | 87.51 | 91.93(+3.36/ -2.87) | 8.37 |
| | LeadFL$^+$ | 86.93 | 89.91(+4.06/-4.33) | 11.99 |
| | EA-PS$^-$ | 86.56 | 89.90(+4.66/ -4.71) | 18.91 |
| | EA-PS | 88.47 | 91.82(+1.11/ -1.08) | 0.68 |
| CTMA | None | 89.72 | 64.64(+4.3 / -3.77) | 16.54 |
| | LDP | 87.4 | 96.37(+1.36 / -1.19) | 3.25 |
| | WBC | 87.79 | 97.93(+0.29 /-0.14) | 0.06 |
| | LeadFL | 87.27 | 87.25(+7.67/ -3.88) | 24.10 |
| | LeadFL$^+$ | 86.62 | 88.01(+4.5/-4.53) | 18.09 |
| | EA-PS$^-$ | 86.04 | 80.88(+7.70/ -9.77) | 47.14 |
| | EA-PS | 86.48 | 81.28(+6.89/ -9.09) | 36.07 |
| multiKrum | None | 89.08 | 64.64(+4.3 /-3.77) | 16.54 |
| | LDP | 86.37 | 31.39(+3.11 / -3.15) | 9.81 |
| | WBC | 86.36 | 31.9(+4.14/-6.19) | 29.81 |
| | LeadFL | 85.94 | 17.38(+6.94 / -7.76) | 28.33 |
| | LeadFL$^+$ | 85.55 | 16.43(+6.65/-4.31) | 20.65 |
| | EA-PS$^-$ | 85.96 | 12.58(+4.96/ -5.11) | 17.4 |
| | EA-PS | 85.41 | 11.79(+2.52/ -2.95) | 4.76 |
| bulyan | None | 88.83 | 69.94(+2.45 / -3.39) | 9.17 |
| | LDP | 85.39 | 27.31(+4.99 / -4.61) | 23.11 |
| | WBC | 85.99 | 32.36(+3.67/-4.89) | 19.42 |
| | LeadFL | 85.3 | 15.14(+9.14/ -11.98) | 61.37 |
| | LeadFL$^+$ | 85.12 | 13.77(+8.39/-7.46) | 59.88 |
| | EA-PS$^-$ | 85.34 | 11.38(+16.85/ -7.63) | 93.83 |
| | EA-PS | 84.97 | 12.45(+6.2/ -3.53) | 22.72 |

Table 4: Comparison under 9-pixel pattern backdoor attack on non-IID FashionMNIST dataset

parameter limiting requires finding a more stable parameter space through optimization.

## F. THE USE OF LARGE LANGUAGE MODELS (LLMs)

In our writing process, the large language model only played the role of correcting grammar and spelling errors.

| Server-side Defense | Client-side Defense | MA | BA | VAR |
|---|---|---|---|---|
| None | None | 89.83 | 96.03(+0.33 / -0.26) | 0.08 |
| | LDP | 88.32 | 86.54(+1.27 / 0.68) | 0.77 |
| | WBC | 88.12 | 86.55(+0.4 / -0.33) | 0.09 |
| | LeadFL | 87.35 | 89.83(+2.1/ -1.73) | 2.49 |
| | LeadFL$^+$ | 86.99 | 86.42(+0.12/-0.86) | 0.68 |
| | EA-PS$^-$ | 86.6 | 86.24(+1.22/ -1.33) | 2.07 |
| | EA-PS | 86.75 | 88.76(+1.26 / -2.49) | 4.67 |
| CMA | None | 89.69 | 91.19(+1.66 /-2.39) | 3.54 |
| | LDP | 87.15 | 94.63(+1.7 /-1.30) | 1.56 |
| | WBC | 86.99 | 95.29(+0.83 / -0.67) | 0.39 |
| | LeadFL | 87.71 | 89.53(+1.54/ -1.94) | 2.76 |
| | LeadFL$^+$ | 87.33 | 86.24(+1.12/-1.66) | 2 |
| | EA-PS$^-$ | 86.86 | 85.48(+3.59/ -2.1) | 6.52 |
| | EA-PS | 86.87 | 84.84(+2.11/ -1.08) | 3.34 |
| CTMA | None | 89.85 | 92.47(+0.36 / -0.6) | 0.2 |
| | LDP | 88.08 | 96.32(+0.75 / -0.8) | 0.53 |
| | WBC | 87.88 | 87.71(+0.7 /-0.44) | 0.37 |
| | LeadFL | 87.17 | 87.71(+0.7/ -0.44) | 0.37 |
| | LeadFL$^+$ | 86.23 | 86.59(+1.32/-2.04) | 5.21 |
| | EA-PS$^-$ | 87.16 | 86.09(+1.53/ -1.54) | 4.69 |
| | EA-PS | 86.78 | 84.06(+4.13/ -8.11) | 49.33 |
| multiKrum | None | 89.62 | 22.6(+8.42 /-5.59) | 33.06 |
| | LDP | 86.89 | 71.45(+3.97 /-4.07) | 11.3 |
| | WBC | 86.7 | 72.18(+4.24 / -5.59) | 18.73 |
| | LeadFL | 86.77 | 60.5(+10.3 / -8.23) | 64.4 |
| | LeadFL$^+$ | 86.03 | 56(+11.06/-7.79) | 93.83 |
| | EA-PS$^-$ | 86.19 | 49.87(+7.45/ -4.67) | 42.56 |
| | EA-PS | 86.32 | 45.32(+4.5/ -3.91) | 17.94 |
| bulyan | None | 89.26 | 73.04(+7.39 /-12.98) | 83.48 |
| | LDP | 85.8 | 70.31(+2.56/-1.6) | 3.21 |
| | WBC | 85.63 | 68.39(+1.63 /-3.63) | 3.48 |
| | LeadFL | 85.63 | 58.72(+9.96/ -21.83) | 215.5 |
| | LeadFL$^+$ | 84.99 | 51.39(+5.28/-3.69) | 41.29 |
| | EA-PS$^-$ | 85.55 | 45.45(+10.56/ -22.91) | 237.83 |
| | EA-PS | 85.57 | 45.68(+9.51/ -15.77) | 189.3 |

Table 5: Comparison under 1-pixel pattern backdoor attack on IID FashionMNIST dataset

| Server-side Defense | Client-side Defense | MA | BA | VAR |
|---|---|---|---|---|
| None | None | 89.8 | 96.2(+0.87 / -0.79) | 0.47 |
| | LDP | 87.95 | 85.01(+0.34 / -0.63) | 0.19 |
| | WBC | 87.76 | 85.39(+1.46 / -1.23) | 1.31 |
| | LeadFL | 87.31 | 88.52(+0.76 / -0.72 ) | 0.54 |
| | LeadFL$^+$ | 86.51 | 86.47(+0.61/-0.88) | 0.55 |
| | EA-PS$^-$ | 86.47 | 86.06(+2.58 / -3.69 ) | 7 |
| | EA-PS | 86.75 | 86.46(+1.96 / -2.76 ) | 6.06 |
| CMA | None | 89.62 | 90.77(+0.58 /-0.57) | 0.33 |
| | LDP | 86.62 | 94.24(+0.53 /-0.50) | 0.19 |
| | WBC | 86.27 | 95.16(+0.61 / -0.59) | 0.35 |
| | LeadFL | 87.55 | 91.27(+1.76 / -2.53 ) | 5.03 |
| | LeadFL$^+$ | 86.77 | 87.89(+1.46/-2.23) | 6.87 |
| | EA-PS$^-$ | 86.57 | 85.52(+2.87 / -1.87 ) | 4.42 |
| | EA-PS | 86.63 | 84.96(+0.59 / -0.35 ) | 0.26 |
| CTMA | None | 89.7 | 92.92(+0.99 / -0.87) | 0.86 |
| | LDP | 87.88 | 95.91(+0.27 / -0.37) | 0.07 |
| | WBC | 88.06 | 96.4(+0.23 /-0.23) | 0.5 |
| | LeadFL | 86.95 | 87.64(+1.98 / -4.73 ) | 9.32 |
| | LeadFL$^+$ | 86.72 | 83.59(+1.29/-3.61) | 7.64 |
| | EA-PS$^-$ | 86.44 | 82.44(+1.15 / -1.68 ) | 1.45 |
| | EA-PS | 86.82 | 83.83(+1.77 / -2.42 ) | 4.72 |
| multiKrum | None | 89.35 | 57.07(+2.62 /-2.87) | 7.57 |
| | LDP | 86.29 | 32.4(+4.26 / -4.77) | 14.36 |
| | WBC | 86.35 | 43.78(+1.32/-1.72) | 1.72 |
| | LeadFL | 86.6 | 47.5(+7.29 / -8.14 ) | 66.29 |
| | LeadFL$^+$ | 86.16 | 40.96(+5.66/-4.68) | 20.36 |
| | EA-PS$^-$ | 85.64 | 34.97(+13.74 / -7.53 ) | 101.25 |
| | EA-PS | 86.03 | 35.27(+5.02 / -3.01 ) | 32.24 |
| bulyan | None | 86.75 | 56.33(+2.18 / -3.9 ) | 11.44 |
| | LDP | 85.49 | 32.25(+5.48 / -3.75) | 23.54 |
| | WBC | 85.55 | 28.02(+1.88/-3.18) | 5.1 |
| | LeadFL | 85.44 | 32.32(+8.4 / -5.7 ) | 46.74 |
| | LeadFL$^+$ | 85.41 | 33.65(+4.41/-2.15) | 26.62 |
| | EA-PS$^-$ | 85.26 | 27.29(+3.79 / -3.95 ) | 29.88 |
| | EA-PS | 84.88 | 22.13(+3.89 / -4.58 ) | 18.3 |

Table 6: Comparison under 1-pixel pattern backdoor attack on non-IID FashionMNIST dataset

| Server-side Defense | Client-side Defense | MA | BA | VAR |
|---|---|---|---|---|
| None | None | 89.27 | 23.48(+0.17 / -0.51) | 0.49 |
| | LDP | 87.81 | 13.45(+0.81 / -0.73) | 0.39 |
| | WBC | 87.84 | 13.86(+2.13 / -0.06) | 2.26 |
| | LeadFL | 87.37 | 16.26(+1.14/ -0.61) | 1.46 |
| | LeadFL$^+$ | 87.16 | 17.21(+1.03/-0.72) | 1.33 |
| | EA-PS$^-$ | 86.88 | 15.44(+2.17/ -1.88) | 3.14 |
| | EA-PS | 86.57 | 15.62(+0.96/ -0.87) | 1.05 |
| CMA | None | 88.01 | 20.16(+3.71/-2.01) | 22.72 |
| | LDP | 86.16 | 28.94(+4.37/-1.01) | 20.38 |
| | WBC | 87.07 | 31.88(+5.83 / -2.05) | 28.81 |
| | LeadFL | 87.51 | 14.35(+1.26/ -0.81) | 1.27 |
| | LeadFL$^+$ | 87.18 | 15.26(+1.13/-1.19) | 1.29 |
| | EA-PS$^-$ | 86.99 | 13.57(+2.05/ -3.41) | 8.38 |
| | EA-PS | 86.79 | 13.39(+0.95/ -1.73) | 3.44 |
| CTMA | None | 87.95 | 24.39(+0.31 / -1.14) | 1.23 |
| | LDP | 87.14 | 39.86(+7.19 / -1.33) | 23.74 |
| | WBC | 87.54 | 38.61(+7.54 /-2.09) | 30.98 |
| | LeadFL | 87.38 | 18.51(+5.31/ -2.96) | 23.68 |
| | LeadFL$^+$ | 87.2 | 17.37(+4.36/-2.58) | 16.84 |
| | EA-PS$^-$ | 87.03 | 15.93(+4.72/ -4.08) | 14.49 |
| | EA-PS | 86.95 | 16.66(+2.19/ -1.41) | 5.87 |
| multiKrum | None | 87.02 | 14.31(+1.31 /-1.35) | 3.64 |
| | LDP | 87.34 | 15.92(+4.28 / -5.87) | 39.11 |
| | WBC | 86.69 | 16.45(+3.22/-6.26) | 27.19 |
| | LeadFL | 86.29 | 10.97(+4.14 / -2.15) | 14.53 |
| | LeadFL$^+$ | 86.34 | 11.86(+2.46/-1.51) | 4.73 |
| | EA-PS$^-$ | 86.07 | 9.89(+3.53/ -3.26) | 13.62 |
| | EA-PS | 85.94 | 10.77(+2.87/ -2.08) | 4.02 |
| bulyan | None | 86.93 | 19.21(+3.53 / -2.61) | 9.66 |
| | LDP | 85.66 | 15.78(+3.27 / -5.65) | 27.88 |
| | WBC | 85.59 | 15.26(+2.26/-5.21) | 19.76 |
| | LeadFL | 86.55 | 12.41(+1.24/ -1.72) | 4.23 |
| | LeadFL$^+$ | 86.25 | 13.05(+0.93/-1.47) | 2.08 |
| | EA-PS$^-$ | 86.49 | 11.8(+1.94/ -2.56) | 5.37 |
| | EA-PS | 86.14 | 12.75(+1.16/ -2.02) | 3.38 |

Table 7: Comparison under Spectrum backdoor attack on IID FashionMNIST dataset

| Server-side Defense | Client-side Defense | MA | BA | VAR |
|---|---|---|---|---|
| None | None | 88.69 | 32.04(+0.12 / -0.73) | 0.81 |
| | LDP | 87.77 | 12.76(+0.41 / -0.36) | 0.08 |
| | WBC | 87.32 | 13.59(+0.34/ -0.27) | 0.06 |
| | LeadFL | 87.5 | 17.91(+2/ -1.81) | 5.38 |
| | LeadFL$^+$ | 87.28 | 17.1(+1.53/-0.77) | 1.33 |
| | EA-PS$^-$ | 86.91 | 16.08(+3.37/ -2.48) | 14.39 |
| | EA-PS | 86.58 | 16.02(+0.89/ -0.77) | 1.14 |
| CMA | None | 88.41 | 25.76(+1.61/-2.98) | 7.03 |
| | LDP | 85.06 | 31.01(+3.1/-3.38) | 29.13 |
| | WBC | 86.12 | 31.4(+3.84 / -2.83) | 23.57 |
| | LeadFL | 87.19 | 15.09(+1.28/-0.7) | 1.26 |
| | LeadFL$^+$ | 87.06 | 14.98(+1.19/-0.95) | 1.41 |
| | EA-PS$^-$ | 86.81 | 12.7(+1.98/ -2.61) | 5.88 |
| | EA-PS | 86.44 | 13.02(+0.78/ -0.82) | 0.74 |
| CTMA | None | 88.67 | 28.27(+0.84 / -0.65) | 0.94 |
| | LDP | 88.1 | 39.17(+5.59 / -3.27) | 24.64 |
| | WBC | 87.82 | 41.8(+5.05 /-4.98) | 40.58 |
| | LeadFL | 87.07 | 19.11(+1.04/ -1.46) | 2.06 |
| | LeadFL$^+$ | 86.92 | 19.87(+1.16/-0.98) | 1.33 |
| | EA-PS$^-$ | 87.02 | 17.61(+2.36/ -1.98) | 6.42 |
| | EA-PS | 86.81 | 17.86(+1.16/ -1.03) | 1.71 |
| multiKrum | None | 86.95 | 16.28(+3.46 /-2.17) | 11.84 |
| | LDP | 86.55 | 14.16(+0.99 / -1.81) | 2.74 |
| | WBC | 86.56 | 10.12(+0.12/-0.28) | 0.05 |
| | LeadFL | 86.36 | 10.53(+0.42 / -0.5) | 0.17 |
| | LeadFL$^+$ | 86.22 | 11.31(+0.77/-1.12) | 0.94 |
| | EA-PS$^-$ | 86.19 | 9.16(+1.15/ -0.76) | 0.64 |
| | EA-PS | 86.01 | 11.26(+0.32/ -0.52) | 0.55 |
| bulyan | None | 86.84 | 17.61(+2.13 / -1.86) | 3.17 |
| | LDP | 85.57 | 10.4(+0.67 / -0.6) | 0.33 |
| | WBC | 85.35 | 10.76(+0.53/-0.84) | 0.26 |
| | LeadFL | 84.55 | 10.21(+0.08 / -0.16) | 0.07 |
| | LeadFL$^+$ | 84.48 | 11.12(+0.41/-0.27) | 0.13 |
| | EA-PS$^-$ | 84.71 | 10.2(+0.91/ -0.69) | 0.78 |
| | EA-PS | 84.18 | 11.59(+0.06/ -0.22) | 0.08 |

Table 8: Comparison under Spectrum backdoor attack on NON-IID FashionMNIST dataset

| | | IID | NON-IID |
|---|---|---|---|
| Server-side Defense | Client-side Defense | MA | MA |
| None | None | 88.4 | 88.33 |
| | LDP | 87.19 | 86.62 |
| | WBC | 86.29 | 87.16 |
| | LeadFL | 85.79 | 86.44 |
| | leadFL$^+$ | 85.26 | 85.46 |
| | EA-PS$^-$ | 86.03 | 86.52 |
| | EA-PS | 85.98 | 85.42 |
| CMA | None | 88.6 | 88.56 |
| | LDP | 83.29 | 84.4 |
| | WBC | 81.49 | 81.26 |
| | LeadFL | 86.24 | 87.37 |
| | leadFL$^+$ | 85.51 | 85.64 |
| | EA-PS$^-$ | 86.62 | 87.81 |
| | EA-PS | 85.69 | 85.71 |
| CTMA | None | 89.12 | 89 |
| | LDP | 86.66 | 85.66 |
| | WBC | 85.99 | 85.41 |
| | LeadFL | 87.75 | 87.14 |
| | leadFL$^+$ | 86.43 | 85.26 |
| | EA-PS$^-$ | 87.61 | 87.37 |
| | EA-PS | 86.8 | 85.31 |
| multiKrum | None | 89.65 | 85.66 |
| | LDP | 86.5 | 85.64 |
| | WBC | 86.65 | 86 |
| | LeadFL | 87 | 86.3 |
| | leadFL$^+$ | 86.28 | 85.41 |
| | EA-PS$^-$ | 87.04 | 86.24 |
| | EA-PS | 86.33 | 85.7 |
| bulyan | None | 87.42 | 86.81 |
| | LDP | 84.37 | 84.76 |
| | WBC | 85.19 | 84.43 |
| | LeadFL | 86.01 | 84.68 |
| | leadFL$^+$ | 85.42 | 84.18 |
| | EA-PS$^-$ | 86.34 | 85.47 |
| | EA-PS | 85.44 | 85.25 |

Table 9: Comparison under Label Flipping attack on IID and NON-IID FashionMNIST dataset

| Server-side Defense | Client-side Defense | MA | BA | VAR |
|---|---|---|---|---|
| None | None | 57.4 | 79.02(+0.74/-1.35) | 1.37 |
| | LDP | 54.21 | 79.61(+1.32/-2.02) | 3.16 |
| | WBC | 53.7 | 78.5(+0.96/-0.53) | 0.7 |
| | LeadFL | 36.41 | 76.9(+7.03/-5.51) | 41.04 |
| | LeadFL$^+$ | 35.82 | 70.97(+4.22/-4.41) | 33.85 |
| | EA-PS$^-$ | 35.01 | 79.35(+4.34/-5.6) | 25.91 |
| | EA-PS | 44.17 | 77.8(+1.95/-2.93) | 6.69 |
| CMA | None | 55.87 | 58.83(+6.21/-7.96) | 52.49 |
| | LDP | 40.25 | 89.11(+1.96/-2.32) | 4.67 |
| | WBC | 38.71 | 87.15(+3.94/-5.54) | 24.39 |
| | LeadFL | 36.45 | 78.34(+3.14/-4.32) | 14.66 |
| | LeadFL$^+$ | 40.56 | 73.61(+2.58/-3.81) | 8.62 |
| | EA-PS$^-$ | 30.75 | 62.05(+4.08/-6.86) | 35.68 |
| | EA-PS | 42.9 | 63.57(+0.05/-0.04) | 0.01 |
| CTMA | None | 56.25 | 64.49(+2.26/-2.09) | 4.76 |
| | LDP | 54.29 | 90.25(+0.74/-1.43) | 1.53 |
| | WBC | 54.08 | 90.54(+2.97/-1.91) | 6.8 |
| | LeadFL | 38.2 | 64.87(+10.51/-8.57) | 93.85 |
| | LeadFL$^+$ | 38.82 | 61.75(+5.39/-7.21) | 56.73 |
| | EA-PS$^-$ | 41.42 | 57.24(+5.33/-7.38) | 43.53 |
| | EA-PS | 45.54 | 65.95(+1.55/-2.65) | 5.35 |
| multiKrum | None | 56.36 | 72.23(+9.34/-7.89) | 75.88 |
| | LDP | 53.15 | 92.02(+0.39/-0.2) | 0.11 |
| | WBC | 51.49 | 89.97(+0.88/-0.64) | 0.62 |
| | LeadFL | 42.06 | 60.63(+11.72/-13.83) | 166.51 |
| | LeadFL$^+$ | 40.29 | 63.39(+1.83/-1.72) | 5.2 |
| | EA-PS$^-$ | 32.12 | 70.03(+9.28/-10.65) | 100.78 |
| | EA-PS | 48 | 62.75(+0.43/-0.44) | 0.38 |
| bulyan | None | 55.87 | 69.29(+9.5/-5.25) | 67.94 |
| | LDP | 49.32 | 90.27(+0.7/-1.32) | 1.3 |
| | WBC | 49.7 | 89.96(+1.14/-1.02) | 1.17 |
| | LeadFL | 37.66 | 79.16(+5.23/-5.24) | 54.75 |
| | LeadFL$^+$ | 38.25 | 78.06(+4.29/-5.97) | 32.56 |
| | EA-PS$^-$ | 30.17 | 66.37(+18.15/-20.75) | 383.42 |
| | EA-PS | 30.25 | 61.3(+0.14/-0.15) | 0.04 |

Table 10: Comparison under 9-pixel pattern backdoor attack on IID CIAFR10 dataset

| Server-side Defense | Client-side Defense | MA | BA | VAR |
|---|---|---|---|---|
| None | None | 55.84 | 80.22(+1.14/-1.28) | 1.48 |
| | LDP | 52.06 | 77.4(+2.5/-2.34) | 5.89 |
| | WBC | 52.56 | 78.96(+2.5/-2.34) | 5.89 |
| | LeadFL | 45.08 | 78.78(+5.14/-6.12) | 32.38 |
| | LeadFL$^+$ | 39.11 | 78(+2.58/-1.73) | 10.52 |
| | EA-PS$^-$ | 36.19 | 78.57(+5.12/-7.18) | 40.99 |
| | EA-PS | 43.22 | 77.1(+1.46/-2.98) | 5.52 |
| CMA | None | 54.35 | 61.76(+4.34/-5.83) | 27.48 |
| | LDP | 39.22 | 85.07(+1.03/-1.25) | 1.34 |
| | WBC | 40.91 | 85.2(+2.2/-1.57) | 3.87 |
| | LeadFL | 32.79 | 61.21(+2.2/-3.83) | 11.06 |
| | LeadFL$^+$ | 36.99 | 59.51(+5.26/-4.18) | 33.69 |
| | EA-PS$^-$ | 32.76 | 63.66(+3.04/-4.93) | 18.54 |
| | EA-PS | 44.47 | 63.21(+0.13/-1.89) | 3.35 |
| CTMA | None | 55.34 | 66.7(+1.42/-2.35) | 4.17 |
| | LDP | 53.29 | 89.64(+1.72/-1.61) | 2.78 |
| | WBC | 53.75 | 92.3(+1.88/-1.81) | 3.4 |
| | LeadFL | 30.2 | 60.9(+18.7/-13.65) | 280.77 |
| | LeadFL$^+$ | 39.25 | 57.17(+4.22/-3.67) | 20.36 |
| | EA-PS$^-$ | 38.91 | 49.53(+5.44/-6.34) | 35.29 |
| | EA-PS | 44.57 | 54.82(+14.84/-9.17) | 189.97 |
| multiKrum | None | 54.97 | 68.5(+1.49/-1.01) | 1.73 |
| | LDP | 51.06 | 92.75(+0.96//-1.38) | 1.49 |
| | WBC | 50.07 | 92.13(+0.94/-0.61) | 0.69 |
| | LeadFL | 29.44 | 58.13(+5.16/-3.85) | 21.63 |
| | LeadFL$^+$ | 30.22 | 61.55(+3.95/-2.48) | 14.22 |
| | EA-PS$^-$ | 38.12 | 53.42(+2.9/-2.8) | 16.76 |
| | EA-PS | 39.81 | 52.2(+2.13/-2.44) | 8.78 |
| bulyan | None | 53.63 | 70.41(+10.4/-5.85) | 81.51 |
| | LDP | 45.64 | 90.42(+1.36/-0.75) | 1.39 |
| | WBC | 46.16 | 89.55(+1.91/-1.14) | 2.79 |
| | LeadFL | 37.3 | 64.65(+5.71/-3.98) | 25.71 |
| | LeadFL$^+$ | 39.28 | 61.76(+3.4/-4.18) | 18.55 |
| | EA-PS$^-$ | 36.35 | 71.11(+5.35/-4.2) | 23.79 |
| | EA-PS | 38.55 | 66.05(+4.42/-3.15) | 15.58 |

Table 11: Comparison under 9-pixel pattern backdoor attack on non-IID CIFAR10 dataset

| Server-side Defense | Client-side Defense | MA | BA | VAR |
|---|---|---|---|---|
| None | None | 55.73 | 39.25(+5.91/-4.37) | 28.22 |
| | LDP | 53.64 | 49.19(+2.06/-3.57) | 9.64 |
| | WBC | 52.77 | 44.23(+8.8//-5.3) | 58.88 |
| | LeadFL | 40.67 | 53.09(+11.21/-13.11) | 158.59 |
| | LeadFL$^+$ | 39.36 | 54.79(+12.71/-10.88) | 141.63 |
| | EA-PS$^-$ | 38.38 | 61.14(+1.92/-2.42) | 4.9 |
| | EA-PS | 39.79 | 54.52(+3.59/-6.83) | 34.95 |
| CMA | None | 54.05 | 21.14(+2.81/-2.56) | 7.26 |
| | LDP | 37.23 | 33.4(+2.21/-3.01) | 7.3 |
| | WBC | 34.32 | 40.3(+5.45/-7.55) | 45.6 |
| | LeadFL | 40.14 | 46.54(+17.35/-17.36) | 301.31 |
| | LeadFL$^+$ | 41.13 | 34.19(+7.31/-4.29) | 48.78 |
| | EA-PS$^-$ | 34.64 | 44.71(+23.3/-16.13) | 427.22 |
| | EA-PS | 39.8 | 32.58(+0.09/-0.1) | 0.02 |
| CTMA | None | 56.54 | 30.95(+0.55/-0.5) | 0.28 |
| | LDP | 54.24 | 72.49(+1.98/-1.69) | 3.44 |
| | WBC | 53.98 | 73.17(+2.25/-1.45) | 3.92 |
| | LeadFL | 49.31 | 64.46(+9/-9.01) | 162.16 |
| | LeadFL$^+$ | 42.56 | 54.58(+7.25/-6.16) | 48.1 |
| | EA-PS$^-$ | 34.07 | 50.36(+28.63/-14.5) | 614.8 |
| | EA-PS | 36.24 | 42.62(+0.49/-0.34) | 0.19 |
| multiKrum | None | 55.71 | 58.08(+2/-3.31) | 8.33 |
| | LDP | 51.81 | 73.06(+3.38/-1.8) | 8.56 |
| | WBC | 52.05 | 76.63(+2.64/-2.08) | 5.8 |
| | LeadFL | 30 | 64.26(+15.5/-29.21) | 640.64 |
| | LeadFL$^+$ | 32.88 | 61.72(+7.98/-5.76) | 62.41 |
| | EA-PS$^-$ | 31.05 | 67.4(+11.3/-11.23) | 126.97 |
| | EA-PS | 40.3 | 52.5(+2.4/-2.7) | 15 |
| bulyan | None | 54.06 | 60.57(+0.77/- 1.02) | 0.85 |
| | LDP | 49.16 | 76.12( +0.89/- 1.42) | 1.53 |
| | WBC | 48.56 | 76.48(+0.71/-0.72) | 1.02 |
| | LeadFL | 34.75 | 63(+16.85/-13.28) | 236.4 |
| | LeadFL$^+$ | 33.76 | 66.14(+8.77/-7.39) | 120.55 |
| | EA-PS$^-$ | 29.37 | 59.15(+21.54/-23.12) | 500.37 |
| | EA-PS | 33.88 | 63.74(+9.04/-10.51) | 97.22 |

Table 12: Comparison under 1-pixel pattern backdoor attack on IID CIFAR10 dataset

| Server-side Defense | Client-side Defense | MA | BA | VAR |
|---|---|---|---|---|
| None | None | 56.17 | 41.3(+8.58/-7.35) | 64.56 |
| | LDP | 52.29 | 49.51(+1.67/-1.67) | 2.78 |
| | WBC | 54.04 | 45.37(+4.17/-5.91) | 27.72 |
| | LeadFL | 37.64 | 51.84(+7.01/-8.62) | 63.03 |
| | LeadFL$^+$ | 38.81 | 58.33(+3.21/-3.7) | 10.55 |
| | EA-PS$^-$ | 37.8 | 54.74(+3.86/-5.12) | 21.37 |
| | EA-PS | 40.31 | 50.6(+3.84/-4.05) | 15.6 |
| CMA | None | 54.79 | 21.31(+0.74/-1.34) | 1.34 |
| | LDP | 37.52 | 37.18(+3.07/-4.94) | 18.7 |
| | WBC | 30.25 | 35.95(+19.66/-13.88) | 306.25 |
| | LeadFL | 37.39 | 31.47(+1.3/- 1.64) | 2.35 |
| | LeadFL$^+$ | 36.2 | 36.46(+3.29/-2.8) | 30.19 |
| | EA-PS$^-$ | 35.98 | 37.66(+9.34/-16.43) | 83.36 |
| | EA-PS | 41.36 | 33.18(+7.08/-10.69) | 91.01 |
| CTMA | None | 55.69 | 29.58(+3.73/-4.44) | 17.04 |
| | LDP | 52.81 | 75.64(+1.37/-2.16) | 3.6 |
| | WBC | 52.75 | 75.24(+1.04/-1.24) | 1.33 |
| | LeadFL | 30.34 | 43.3(+16.8/-9.77) | 213.57 |
| | LeadFL$^+$ | 40.5 | 42.61(+4.71/-3.92) | 40.6 |
| | EA-PS$^-$ | 38.96 | 39.6(+8.29/-9.54) | 57.13 |
| | EA-PS | 43.96 | 40.01(+3.46/-2.09) | 9.12 |
| multiKrum | None | 53.81 | 47.25(+6.4/-3.99) | 31.3 |
| | LDP | 49.94 | 75.55(+3.72/-3.44) | 12.86 |
| | WBC | 49.54 | 76.56(+2.66/-4.84) | 17.64 |
| | LeadFL | 30.23 | 61.74(+3.92/-2.33) | 11.71 |
| | LeadFL$^+$ | 33.4 | 56.84(+3.89/-3.72) | 26.28 |
| | EA-PS$^-$ | 36.63 | 45.2(+2.41/-1.61) | 4.54 |
| | EA-PS | 31.33 | 52.5(+2.49/-4.46) | 15 |
| bulyan | None | 55.54 | 56.28(+4.93/-6.65) | 35.72 |
| | LDP | 44.54 | 79.19(+0.86/-0.78) | 0.68 |
| | WBC | 44.45 | 78.18(+2.79/-2.94) | 8.22 |
| | LeadFL | 32.31 | 65.96(+9.02/-7.79) | 71.8 |
| | LeadFL$^+$ | 39.92 | 56.02(+7.29/-8.06) | 92.72 |
| | EA-PS$^-$ | 40.5 | 58.19(+1.59/-1.65) | 2.63 |
| | EA-PS | 43.12 | 59.11(+1.78/-3.18) | 7.61 |

Table 13: Comparison under 1-pixel pattern backdoor attack on non-IID CIFAR10 dataset

| Server-side Defense | Client-side Defense | MA | BA | VAR |
|---|---|---|---|---|
| None | None | 57.42 | 67.82(+5.34/-4.91) | 29.51 |
| | LDP | 53.36 | 64.03(+1.26/-1.74) | 3.82 |
| | WBC | 55.16 | 48.27(+4.15/-4.74) | 15.79 |
| | LeadFL | 34.05 | 56.73(+3.71/-4.19) | 16.28 |
| | LeadFL$^+$ | 33.81 | 57.44(+2.37/-1.96) | 7.82 |
| | EA-PS$^-$ | 33.77 | 47.48(+4.62/-5.71) | 27.06 |
| | EA-PS | 34.76 | 46.55(+2.17/-1.7) | 3.83 |
| CMA | None | 53.17 | 49.28(+2.74/-2.28) | 8.51 |
| | LDP | 39.83 | 77.65(+8.07/-6.38) | 75.46 |
| | WBC | 39.48 | 73.92(+4.31/-2.4) | 29.06 |
| | LeadFL | 31.15 | 45.78(+2.71/-2.39) | 13.68 |
| | LeadFL$^+$ | 30.78 | 46.72(+1.74/-3.41) | 23.29 |
| | EA-PS$^-$ | 32.61 | 43.19(+4.03/-3.91) | 57.28 |
| | EA-PS | 35.84 | 42.43(+1.63/-1.16) | 2.14 |
| CTMA | None | 55.17 | 61.72(+3.12/-3.81) | 22.14 |
| | LDP | 48.79 | 79.23(+2.74/-2.83) | 19.47 |
| | WBC | 47.61 | 81.26(+5.34/-6.59) | 63.85 |
| | LeadFL | 31.21 | 49.71(+8.46/-15.3) | 244.86 |
| | LeadFL$^+$ | 32.72 | 47.28(+3.85/-2.18) | 22.71 |
| | EA-PS$^-$ | 32.15 | 43.89(+4.37/-5.15) | 63.87 |
| | EA-PS | 32.9 | 44.05(+4.32/-4.11) | 11.87 |
| multiKrum | None | 53.78 | 76.94(+0.31/-0.48) | 0.39 |
| | LDP | 42.85 | 85.04(+5.24/-6.82) | 47.72 |
| | WBC | 45.97 | 88.46(+4.38/-2.75) | 25.09 |
| | LeadFL | 33.95 | 76.77(+9.21/-7.18) | 105.1 |
| | LeadFL$^+$ | 32.64 | 77.31(+3.06/-5.17) | 42.75 |
| | EA-PS$^-$ | 33.79 | 74.66(+3.19/-1.64) | 14.63 |
| | EA-PS | 32.41 | 73.06(+4.59/-3.72) | 25.26 |
| bulyan | None | 46.02 | 52.97(+5.41/-6.24) | 23.38 |
| | LDP | 42.74 | 71.09(+6.96/-4.28) | 17.59 |
| | WBC | 41.38 | 74.13(+3.74/-6.27) | 32.83 |
| | LeadFL | 33.44 | 40.85(+7.36/-4.27) | 52.83 |
| | LeadFL$^+$ | 33.08 | 42.69(+2.14/-3.23) | 10.75 |
| | EA-PS$^-$ | 33.15 | 47.97(+3.17/-5.68) | 28.74 |
| | EA-PS | 32.18 | 49.76(+2.86/-3.67) | 12.17 |

Table 14: Comparison under Spectrum backdoor attack on IID CIFAR10 dataset

| Server-side Defense | Client-side Defense | MA | BA | VAR |
|---|---|---|---|---|
| None | None | 53.5 | 66.31(+3.31/-3.63) | 14.65 |
| | LDP | 50.07 | 65.06(+3.17/-2.49) | 15.58 |
| | WBC | 48.98 | 65.76(+5.27/-3.18) | 29.41 |
| | LeadFL | 30.18 | 85.02(+6.73/-3.27) | 84.48 |
| | LeadFL$^+$ | 29.78 | 42.04(+5.59/-5.73) | 78.68 |
| | EA-PS$^-$ | 31.6 | 42.74(+3.65/-5.49) | 62.85 |
| | EA-PS | 31.04 | 47.57(+3.2/-6.37) | 77.29 |
| CMA | None | 52.06 | 51.06(+2.18/-1.96) | 3.62 |
| | LDP | 33.59 | 89.56(+0.87/-2.41) | 4.94 |
| | WBC | 38.17 | 77.79(+2.16/-3.74) | 19.05 |
| | LeadFL | 28.01 | 33.83(+8.36/-7.05) | 45.05 |
| | LeadFL$^+$ | 34.63 | 43.36(+4.28/-3.89) | 27.99 |
| | EA-PS$^-$ | 32.58 | 37.9(+2.43/-4.97) | 37.52 |
| | EA-PS | 31.79 | 41.02(+2.17/-1.83) | 6.87 |
| CTMA | None | 53.69 | 65.93(+2.2/-2.86) | 12.74 |
| | LDP | 49.02 | 80.08(+1.36/-3.77) | 9.33 |
| | WBC | 48.92 | 82.45(+3.17/-4.23) | 25.62 |
| | LeadFL | 29.65 | 57.59(+6.01/-8.41) | 84.28 |
| | LeadFL$^+$ | 30.12 | 57.22(+3.71/-2.66) | 14.63 |
| | EA-PS$^-$ | 30.05 | 55.46(+4.29/-6.57) | 51.83 |
| | EA-PS | 27.3 | 56.61(+4.62/-6.75) | 37.43 |
| multiKrum | None | 53.25 | 72.86(+1.64/-2.19) | 3.28 |
| | LDP | 45.71 | 87.78(+2.82/-3.59) | 14.01 |
| | WBC | 46.92 | 89.98(+4.27/-3.13) | 29.92 |
| | LeadFL | 33.95 | 55.42(+6.38/-4.61) | 72.54 |
| | LeadFL$^+$ | 37.72 | 49.12(+3.28/-5.85) | 36.09 |
| | EA-PS$^-$ | 35.6 | 45.92(+4.89/-5.13) | 69.23 |
| | EA-PS | 33.75 | 46.12(+3.17/-4.67) | 37.11 |
| bulyan | None | 49.16 | 56.72(+4.36/-5.81) | 29.75 |
| | LDP | 44.48 | 85.15(+1.76/-1.57) | 15.9 |
| | WBC | 43.6 | 89.35(+2.13/-1.77) | 15.72 |
| | LeadFL | 32.1 | 41.33(+3.19/-1.58) | 7.62 |
| | LeadFL$^+$ | 37.05 | 68.88(+1.65/-3.07) | 6.23 |
| | EA-PS$^-$ | 34.1 | 39.79(+7.27/6.09) | 62.97 |
| | EA-PS | 34.81 | 40.13(+1.74/-3.25) | 7.81 |

Table 15: Comparison under Spectrum backdoor attack on NON-IID CIFAR10 dataset

| Server-side Defense | Client-side Defense | IID MA | NON-IID MA |
|---|---|---|---|
| None | None | 53.18 | 53.68 |
| | LDP | 48.29 | 48.53 |
| | WBC | 50.89 | 50.35 |
| | LeadFL | 16.01 | 16.16 |
| | leadFL$^+$ | 22.20 | 25.54 |
| | EA-PS$^-$ | 23.64 | 25.77 |
| | EA-PS | 21.93 | 24.58 |
| CMA | None | 50.87 | 49.67 |
| | LDP | 15.05 | 14.42 |
| | WBC | 14.92 | 14.16 |
| | LeadFL | 15.15 | 18.71 |
| | leadFL$^+$ | 24.14 | 26.69 |
| | EA-PS$^-$ | 18.64 | 19.79 |
| | EA-PS | 16.92 | 18.57 |
| CTMA | None | 52.42 | 51.09 |
| | LDP | 45.67 | 44.28 |
| | WBC | 45.28 | 44.13 |
| | LeadFL | 21.47 | 12.24 |
| | leadFL$^+$ | 22.51 | 19.82 |
| | EA-PS$^-$ | 21.19 | 20.03 |
| | EA-PS | 19.85 | 17.22 |
| multiKrum | None | 52.93 | 51.48 |
| | LDP | 42.1 | 41.44 |
| | WBC | 44.08 | 42.64 |
| | LeadFL | 10.9 | 13.42 |
| | leadFL$^+$ | 19.36 | 13.35 |
| | EA-PS$^-$ | 20.64 | 13.39 |
| | EA-PS | 13.39 | 15.9 |
| bulyan | None | 51.3 | 51.29 |
| | LDP | 40.45 | 34.19 |
| | WBC | 42.43 | 37.78 |
| | LeadFL | 14.27 | 18.94 |
| | leadFL$^+$ | 13.51 | 13.40 |
| | EA-PS$^-$ | 15.85 | 16.27 |
| | EA-PS | 14.8 | 16.08 |

Table 16: Comparison under Label Flipping attack on IID and NON-IID CIFAR10 dataset

| (mean(%)\var($10^{-4}$)) | | IID | | | | | non-IID | | | | |
|---|---|---|---|---|---|---|---|---|---|---|---|
| | $\beta$ | 0.01 | 0.03 | 0.05 | 0.07 | 0.09 | 0.01 | 0.03 | 0.05 | 0.07 | 0.09 |
| 9-pixel | none | 90.24/0.05 | **87.08/6.17** | 95.87/0.61 | **88.7/4.36** | 89.38/10.78 | 87.23/0.44 | **86.47/3.69** | **87.79/0.21** | 87.97/7.68 | 86.96/2.31 |
| | Multikrum | 32.45/51.18 | 25.81/39.85 | **22.38/84.09** | 18.29/0.97 | 24.13/30.63 | 16.17/251.97 | **6.61/8.09** | **7.39/13.4** | 17.55/1.42 | 24.13/30.63 |
| | Bulyan | 21.4/29.19 | 29.25/4.77 | 33.15/175.49 | **19.58/0.01** | 16.24/59.42 | 11.39/10.38 | 6.85/26.68 | **5.19/4.25** | 13.01/48.52 | **6.3/4.24** |
| 1-pixel | none | 91.96/0.12 | 91.90/0.92 | **89.09/0.07** | 87.28/0.44 | 91.60/3.1 | 89.36/19.39 | **86.46/6.06** | 89.35/0.98 | 89.81/0.94 | **87.21/1.85** |
| | Multikrum | **45.58/0.66** | 59.74/176.35 | **44.65/6.67** | 57.04/118.49 | 46.07/74.95 | 39.50/0.64 | **31.59/98.35** | **36.42/132.46** | 51.47/64.03 | 40.69/41.37 |
| | Bulyan | **45.68/189.3** | 54.14/32.44 | **39.62/14.76** | 47.81/184.62 | 55.9/116.15 | 33.06/32.33 | **26.52/0.06** | 30.72/56.31 | **25.40/221.80** | 27.95/1.29 |

Table 17: Comparison of different $\beta$ on FashionMNIST with IID/non-IID settings under 1/9-pixel backdoor attack.

| $\alpha$(mean(%)/var($10^{-4}$)) | | server-defense | 0.1 | 0.2 | 0.3 | 0.4 | 0.5 | 0.6 | 0.7 | 0.8 | 0.9 |
|---|---|---|---|---|---|---|---|---|---|---|---|
| IID | 9-pixel | MultiKrum | 8.85/1.93 | 23/3.96 | 35.74/1.77 | 28.46/10.25 | 13.05/1.32 | 27.73/12.65 | 30.61/1.55 | 25.86/10.09 | 19.75/5.06 |
| | | Bulyan | 26.87/76.91 | 7.96/1.48 | 32.39/59.97 | 42.03/139.27 | 36.81/68.19 | 41.97/136.82 | 33.44/0.33 | 47.14/257.44 | 67.64/37.09 |
| | 1-pixel | MultiKrum | 69.27/15.19 | 43.07/52.73 | 42.03/50.85 | 33.58/5.43 | 37.05/2.83 | 39.29/7.25 | 48.45/4.44 | 38.64/3.88 | 48.12/3.25 |
| | | Bulyan | 31.26/183.19 | 54.07/125.02 | 36.05/145.32 | 34.16/97.46 | 41.12/65.35 | 51.71/95.86 | 73.17/143.11 | 80.37/25.9 | 85.27/0.12 |
| non-IID | 9-pixel | MultiKrum | 5.34/15.48 | 23.33/8.24 | 8.82/3.22 | 10.34/7.22 | 10.49/20.36 | 6.83/14.09 | 10.34/9.32 | 13.34/4.65 | 16.89/12.81 |
| | | Bulyan | 20.14/14.45 | 20.04/21.26 | 7.32/1.14 | 25.5/1.16 | 20.33/6.58 | 19.27/41.37 | 25.57/5.44 | 28.53/4.66 | 30.45/2.12 |
| | 1-pixel | MultiKrum | 22.39/19.34 | 25.39/16.47 | 39.45/10.26 | 36.84/1.55 | 40.07/7.45 | 36.57/2.15 | 42.38/0.01 | 26.67/9.8 | 42.07/13.23 |
| | | Bulyan | 34.66/30.25 | 42.63/275.9 | 18.99/89.01 | 32.57/5.43 | 30.04/31.08 | 30.51/29.67 | 36.45/18.98 | 41.44/28.32 | 42.63/35.36 |

Table 18: Comparison of different $\alpha$ on FashionMNIST with IID/non-IID settings under 1/9-pixel backdoor attack.

| (mean(%)/var(10⁻⁴)) | | IID | | | | | non-IID | | | | |
|---|---|---|---|---|---|---|---|---|---|---|---|
| linear ratio in $\lambda$ | | 0.1 | 0.3 | 0.5 | 0.7 | 0.9 | 0.1 | 0.3 | 0.5 | 0.7 | 0.9 |
| 9-pixel | MultiKrum | 45.02/8.3 | 32.75/8.33 | 10.83/18.66 | 22.86/12.18 | 36.58/16.27 | 16.8/6.78 | 7.81/0.55 | 8.32/6.44 | 10.54/4.81 | 11.73/11.93 |
| | Bulyan | 21.14/13.68 | 15.94/1.08 | 17.67/1.55 | 19.47/10.61 | 27.96/30.29 | 4.81/0.04 | 20.02/14.3 | 7.59/4.32 | 24.47/22.88 | 10.29/7.51 |
| 1-pixel | MultiKrum | 49.96/5.76 | 61.99/7.16 | 47.25/2.54 | 40.34/1.38 | 49.06/1.01 | 54.43/3.83 | 36.39/5.05 | 36.86/1.57 | 39.96/4.42 | 27.84/3.41 |
| | Bulyan | 30.64/1.47 | 46.92/13.68 | 55.78/5.51 | 31/0.46 | 42.94/3.37 | 46.85/1.72 | 30.7/2.74 | 29.88/9.24 | 41.56/4.38 | 28.34/2.4 |

Table 19: Comparison of different linear ratio in $\lambda$ on FashionMNIST with IID/non-IID settings under 1/9-pixel backdoor attack.

| mean%/var10⁻⁴ | | IID | | | | | non-IID | | | | |
|---|---|---|---|---|---|---|---|---|---|---|---|
| $\gamma$ | | 0.01 | 0.03 | 0.05 | 0.07 | 0.09 | 0.01 | 0.03 | 0.05 | 0.07 | 0.09 |
| 9-pixel | MultiKrum | 22.19/17.57 | 42.16/4.42 | 34.44/9.69 | 22.94/1.98 | 27.09/9.35 | 13.49/4.82 | 7.39/2.43 | 28.7/7.3 | 4.39/0.64 | 15.59/8.01 |
| | Bulyan | 34.01/67.96 | 4.55/0.04 | 19.17/31.21 | 10.06/6.71 | 37.61/4.43 | 7.21/0.8 | 7.8/3.24 | 4.47/1.22 | 4.25/0.65 | 9.57/1.02 |
| 1-pixel | MultiKrum | 50.89/0.79 | 40.24/2.11 | 48.95/0.78 | 46.17/7.74 | 48.96/7.2 | 34.32/5.37 | 24.37/0.98 | 27.99/7.66 | 56.77/6.3 | 20.44/7.25 |
| | Bulyan | 43.74/2.29 | 36.78/0.83 | 44.72/17.42 | 40.9/7.87 | 23.76/2 | 29.57/4.60 | 17.61/2.1 | 25.85/9.09 | 25.06/7.05 | 44.9/3.68 |

Table 20: Comparison of different $\gamma$ on FashionMNIST with IID/non-IID settings under 1/9-pixel backdoor attack.

| Time(s/r) | CIFAR10 | | | | FEMNIST | | | |
|---|---|---|---|---|---|---|---|---|
| Defense | FedAvg | multiKrum | Bulyan | CRFL | FedAvg | multiKrum | Bulyan | CRFL |
| LeadFL | 27.98 | 28.66 | 28.21 | 45.13 | 80.75 | 82.63 | 78.89 | 118.51 |
| EA-PS | 22.55 | 23.11 | 22.83 | 40.62 | 56.51 | 58.74 | 58.02 | 109.36 |
| FL-WBC | 20.95 | 21.34 | 21.27 | 38.32 | 44.17 | 50.44 | 49.38 | 98.27 |
| NULL | 19.74 | 20.94 | 20.61 | 37.68 | 43.17 | 47.55 | 46.57 | 97.46 |

Table 21: Overhead analysis

