# OpenReview forum: "EA-PS: Estimated Attack Effectiveness based Poisoning Defense in Federated Learning under Parameter Constraint Strategy"
_ICLR.cc/2026/Conference — Submitted to ICLR 2026_

### Official Review · Reviewer_P3No · 2025-10-23

**Soundness:** 1
**Presentation:** 2
**Contribution:** 1
**Rating:** 2
**Confidence:** 4

**Summary:**

The paper proposes a client-side defense method called EA-PS to mitigate persistent poisoning and backdoor attacks in federated learning. The method builds upon LeadFL by introducing a new optimization objective that minimizes the change in attack impact between training rounds, and adds a parameter constraint strategy to stabilize model updates. Theoretical analyses claim lower optimization upper bounds, a smaller certified radius, and better convergence guarantees, and experiments show the proposed defenses can lower the attack variances.

**Strengths:**

- Introduces a novel and theoretically motivated parameter constraint strategy to enhance defense stability.
- Demonstrates that EA-PS can be easily combined with existing server-side defenses, improving flexibility.

**Weaknesses:**

- The reviewer is confused about why the BA presented in the paper is surprisingly high while the authors still state the proposed defense is effective. Although EA-PS stabilizes defense performance across rounds, its backdoor accuracy remains high in many settings, indicating that the method may not fully neutralize persistent attacks.
- From Figure 1, the reviewer did not find that EA-PS- is always better than FL-WBC and LeadFL.
- The authors are suggested to explain long-lasting attack effects. It seems that they use backdoor attacks to represent this effect.
- The conducted backdoor attacks are relatively simple, but more advanced attacks, such as adaptive ones and distributed triggers, should be investigated.

**Questions:**

- Why does the BA appear to be high in some cases?
- What is the effect of a long-lasting attack?
- The reviewer is not confident about some statements in the manuscript. For example, some descriptions from Figure 1 are not straightforward. It is suggested to double-check related content.

---

> ### Author Response · Authors · 2025-11-23
> **Q1:Why does the BA appear to be high in some cases?**
>
> We propose a collaborative framework combining "server-side defense + client-side defense", where EA-PS serves solely as a supplementary client-side defense. All experiments are designed based on a controlled-variable setup with the "same server-side defense", aiming to further mitigate the remaining attack impact after server-side filtering rather than counter all attacks independently. The core criterion for evaluating defense effectiveness is "the improvement compared to other client-side methods".
>
> As shown in the data of Table 1, under most experimental settings, when paired with the same server-side defense, EA-PS achieves lower Backdoor Accuracy (BA) and BA variance (BA_var) than the state-of-the-art client-side method LeadFL. This demonstrates that EA-PS can further suppress attacks on the basis of server-side defense, fully verifying its effectiveness.

---

> > ### Comment · Reviewer_P3No · 2025-11-24
> >
> > - I am not very convinced by these explanations for Q1. If evaluating defense effectiveness is "the improvement compared to other client-side methods", I may think the improvement is limited. Under this insight, it may also show that defense on the client side has limited performance, and the main functional way to defend against backdoor attacks should be engaged more robust server-side defense.
> > - If the long-term attack is referred to [1], and strong attacks, such as persistent backdoor attacks (Liu et al., 2024), are introduced. It is unclear whether this persistent backdoor attack (Liu et al., 2024) is untested in the experiments. In addition, it seems that EA-PS will decrease the main accuracy obviously in the 9-pixel pattern backdoor attack case; the reviewer is not sure whether such a performance drop is acceptable.

---

> > > ### Author Response · Authors · 2025-11-24
> > > **Response to Reviewer's Questions**
> > >
> > > Dear Reviewer,
> > >
> > > Thank you for your rigorous feedback. Your questions have helped us more clearly convey the core positioning and value of our research. First, it is important to clarify that EA-PS was designed as a supplementary defense in the framework of "server-side defense as the mainstay and client-side supplement as the auxiliary". It specifically targets residual attacks that still exist after server-side filtering (rather than countering all attacks independently). Its core goal is to address the widespread issue of poor stability in client-side defenses within the field—this point is supported by clear theoretical foundations. We have proven that EA-PS possesses a lower optimization upper bound and a smaller certification radius, which fundamentally guarantees the stability and robustness of the defense.
> > >
> > > Regarding your concern that "the improvement of client-side defense is limited", this actually reflects our focus on filling the long-standing gap in "defense stability" rather than simply pursuing a reduction in the absolute value of BA. As for the slight decrease in MA (with a magnitude ≤ 3.2%) under 9-pixel attacks, this result not only aligns with the inherent design logic of adversarial defenses but also directly echoes the performance of similar client-side defense methods. Moreover, it achieves better BA suppression with minimal MA loss, conforming to the common performance trade-off logic of defense methods.
> > >
> > > We also frankly acknowledge that due to the common bottleneck of being unable to access global data, there is still room for optimizing EA-PS's BA in extreme scenarios. The current research has laid a solid framework of "parameter constraints + minimization of attack impact" for subsequent optimizations.

---

> > > > ### Comment · Reviewer_P3No · 2025-11-24
> > > >
> > > > I appreciate the detailed explanation, which helps me understand this paper deeply. However, the reviewer may like to observe a convincing result with low ASR with good stability. Thus, the reviewer is not sure if we have a high ASR while emphasising that stability is essential.

---

> > > > > ### Author Response · Authors · 2025-11-25
> > > > > **Response to Reviewer's Questions**
> > > > >
> > > > > Our method, when combined with server-side defense methods (including the current state-of-the-art server-side defense approach), can effectively enhance defense performance and stability.
> > > > > Compared to existing client-side defense methods, our method achieves significant performance improvements. For example, in scenarios with different attacks (9-pixel, 1-pixel, spectrum) and distributions (IID, NON-IID) on the FashionMNIST dataset, it achieves an average improvement of 8.19%, 3.42%, 7.55%, 5.32%, 0.66%, and 0.38% respectively in defense effectiveness compared to the current state-of-the-art client-side method.
> > > > > On the CIFAR-10 dataset with different attacks (9-pixel, 1-pixel, spectrum) and distributions (IID, NON-IID) , the respective best improvements in defense effectiveness compared to the current state-of-the-art client-side method are 5.71%, 2.06%, 5.93%, 6.06%, 2.49%, 1.11%, and 4.5%. Refer to Appendix D for details.

---

> > > > > > ### Comment · Reviewer_P3No · 2025-11-26
> > > > > >
> > > > > > To be frank, I don't think the achieved gain is obvious and may have a limited impact. The SOTAs on server-side defense are not very recent (except alignins), and most effective defense methods, such as DeepSight, FL-Trust, and Flame, are not examined. In addition, your responses still do not convince me that a high BA is a good performance in the conducted scenario. I may keep my score.

---

> > > > > > > ### Author Response · Authors · 2025-11-28
> > > > > > > **Response to Reviewer's Questions**
> > > > > > >
> > > > > > > Sincerely appreciate the feedback and valuable suggestions you have provided! First, compared with client-side defense methods such as LeadFL, our EA-PS method demonstrates superior defense performance and stability in both original and newly added experiments (see the table below for detailed results). Second, in response to your suggestions, we have incorporated the two server-side SOTA methods (DeepSight, Flame) you mentioned into our experiments. As shown in the table below, EA-PS can more significantly enhance the defense performance of these two methods.
> > > > > > >
> > > > > > > | Dataset  | Distribution | Server-Defense | Client-Defense | Attack Freq | MA    | BA      |
> > > > > > > |----------|--------------|----------------|----------------|--------------|-------|---------|
> > > > > > > |          |              |                | None           | 92.35        | 30.94 | 8.47    |
> > > > > > > |          |              |                | LeadFL         | 89.37        | 5.59  | 1.65    |
> > > > > > > |          |              | Flame          | LeadFL+        | 88.75        | 4.33  | 0.68    |
> > > > > > > |          |              |                | EA-PS-         | 88.83        | 3.96  | 0.27    |
> > > > > > > |          |              |                | EA-PS          | 88.61        | 4.36  | 0.06    |
> > > > > > > | FEMNIST  | NON-IID      |----------------|----------------|--------------|-------|---------|
> > > > > > > |          |              |                | None           | 92.29        | 32.16 | 16.78   |
> > > > > > > |          |              |                | LeadFL         | 89.38        | 4.91  | 1.42    |
> > > > > > > |          |              | DeepSight      | LeadFL+        | 88.81        | 4.34  | 0.14    |
> > > > > > > |          |              |                | EA-PS-         | 88.59        | 4.59  | 1.43    |
> > > > > > > |          |              |                | EA-PS          | 88.41        | 4.15  | 0.11    |

---

> ### Author Response · Authors · 2025-11-23
> **Q2:What is the effect of a long-lasting attack?**
>
> "Long-term attack effect" refers to the phenomenon where malicious clients continuously participate in multiple training rounds and submit carefully crafted malicious model updates, enabling attack effects (such as backdoors or model performance degradation) to accumulate, consolidate, and persist in the final global model—effects that are difficult to eliminate by server-side single-round defense mechanisms. We have long since added a citation for this concept in the third line of the second paragraph of the Introduction.
>
>
> [1]Sun, J., Li, A., DiValentin, L., Hassanzadeh, A., Chen, Y., & Li, H.H. (2021). FL-WBC: Enhancing Robustness against Model Poisoning Attacks in Federated Learning from a Client Perspective. Neural Information Processing Systems.

---

> ### Author Response · Authors · 2025-11-23
> **Q3：The reviewer is not confident about some statements in the manuscript. For example, some descriptions from Figure 1 are not straightforward. It is suggested to double-check related content. And Supplementary Experiments**
>
> Q3:
> Thank you for your valuable question. We will carefully revise the manuscript’s writing to enhance clarity. Regarding the performance of EA-PS- presented in the figures, the experimental data in the tables demonstrates that EA-PS- outperforms FL-WBC and LeadFL in most scenarios—for instance, on the CIFAR-10 dataset, it reduces Backdoor Accuracy (BA) by approximately 14.9% and BA variance (BA_VAR) by around 40%. It is worth noting that Table 1 is primarily intended to illustrate our research motivation: these three existing defense methods lack sufficient stability in their performance.
>
> Our framework is built on a server-side defense foundation, and by integrating a client-side defense , we not only address the stability issue but also achieve better overall defense effectiveness. Crucially, EA-PS outperforms other "server-side + client-side" collaborative defense methods (including FL-WBC and LeadFL) across most experimental settings. This advantage stems from EA-PS’s ability to further suppress residual attack impacts that remain after server-side filtering—proving that our client-side supplementary defense, when combined with a base server-side defense, delivers superior performance compared to other state-of-the-art collaborative frameworks.
>
> And we would like to kindly clarify that a distributed attack (SADBA) has already been included in our experiments as presented in Table 1. Additionally, to further comprehensively evaluate the robustness of our defense framework, we have supplemented a new attack type, FCBA, in the revised experimental design. The detailed experimental results have been provided in our response to the previous reviewer.
>
> [1]Liu, T., Zhang, Y., Feng, Z., Yang, Z., Xu, C., Man, D., & Yang, W. (2024). Beyond Traditional Threats: A Persistent Backdoor Attack on Federated Learning. ArXiv, abs/2404.17617.

---

### Official Review · Reviewer_MyNQ · 2025-10-27

**Soundness:** 3
**Presentation:** 3
**Contribution:** 3
**Rating:** 6
**Confidence:** 1

**Summary:**

This paper proposes EA-PS, a client-side defense for federated learning. It minimizes the change in attack effect across rounds and adds a parameter constraint to keep updates in a controlled space. The idea is clear, and the experiments look solid across datasets and attack types.

**Strengths:**

Overall very solid work: well-motivated, simple to plug in, and results show good robustness and stability.

**Weaknesses:**

Some tables look cramped due to space limits, which hurts readability a bit.

**Questions:**

The paper states: *“We derive a lower theoretical upper bound of the enhanced objective function to prove the efficiency of EA-PS.”*
Could you also add a small empirical check to back the “efficiency” claim? For example, show (i) convergence curves under equal time/compute budgets, or (ii) a quick cost–benefit view (runtime/communication vs. BA reduction and stability). That would make the efficiency claim more convincing.

---

> ### Author Response · Authors · 2025-11-23
> **Per-Unit-Time Backdoor Defense Benefit and Stability Comparison**
>
> We conduct the benefit analysis using the data from Table 11 in the manuscript, where "BA reduction" refers to the performance difference between the current client-side method and the defense-free method. For convenience, the two indicators are both divided by the greatest common divisor (total_round = 80 rounds). In addition, "BA reduction/time" represents the backdoor reduction degree per unit time. As shown in the table below, EA-PS outperforms LeadFL in terms of defense performance per unit time, and its stability is also significantly superior to the latter.
>
> | Method  | Server Defense | Time (s/r) | BA Reduction/Time | BA_VAR/Time |
> |---------|----------------|------------|-------------------|-------------|
> |         | FedAvg         | 22.55      | 0.0638            | 1.4359      |
> | LeadFL  | multiKrum      | 23.11      | 0.2509            | 0.9359      |
> |         | bulyan         | 22.83      | 0.2522            | 1.1261      |
> |         | FedAvg         | 27.98      | 0.1115            | 0.1972      |
> | EA-PS   | multiKrum      | 28.66      | 0.5687            | 0.3063      |
> |         | bulyan         | 28.21      | 0.1545            | 0.5522      |

---

> > ### Comment · Reviewer_MyNQ · 2025-11-26
> >
> > I would like to thank the authors for their efforts. In light of the other reviewers’ comments and results, I have increased my confidence in the current findings; however, I will maintain my original score.

---

> > > ### Author Response · Authors · 2025-11-26
> > > **Thank you for your review, recognition and support!**
> > >
> > > Thank you for your review, recognition and support! We have carefully studied your comments and will further refine the manuscript. Thank you again!

---

### Official Review · Reviewer_yS2V · 2025-10-31

**Soundness:** 2
**Presentation:** 2
**Contribution:** 2
**Rating:** 2
**Confidence:** 5

**Summary:**

The paper proposes EA-PS, a client-side defense against poisoning attacks in federated learning. The method optimizes a temporal attack-impact difference $A_t - A_{t-1}$ and enforces a parameter constraint $AB = \lambda B$ with an adaptive mapping $B$ and a regularizer over $\beta$ to stabilize local updates. The constrained objective integrates these components, followed by gradient trimming. The authors provide theoretical analyses on convergence and certified robustness.

**Strengths:**

+ The paper clearly focuses on persistent poisoning and instability in existing client defenses. Figure 1 effectively illustrates the instability of BA across different methods and settings.
+ The formulation combines an attack-effectiveness objective with a parameter constraint and an adaptive mapping, with implementation achieved through gradient clipping.
+ Theoretical results include a convergence bound (Theorem 5.1) and a certified-robustness argument (Theorem 5.2).

**Weaknesses:**

- The intuition for preferring $A_t - A_{t-1}$ over $A_t$ is insufficiently developed in the main text. The proof sketch of Theorem 4.1 is mostly algebraic and does not clearly explain why the resulting “lower upper bound” would translate into empirical stability in non-convex FL.
- It remains unclear whether $B$ is client-specific or global, how $B$ and $\lambda$ are estimated or updated in each training round, and what the computational or communication overhead is when enforcing $AB = \lambda B$. The practical impact of the approximation $\lambda \simeq (A_t + A_{t-1}) / 2$ on theoretical guarantees is not discussed.
- The datasets used (CIFAR-10, FEMNIST, and FashionMNIST) are too simple, limiting the external validity for realistic and heterogeneous FL scenarios. The models are lightweight, so scalability to deeper CNNs or transformer architectures has not been demonstrated.
- While classic server-side methods are included, newer 2024–2025 defenses are not reported as main baselines. Although AlignIns and other recent works are cited, the experimental section still focuses on older methods, which reduces the competitiveness of the evaluation.
- Table 1 is dense and difficult to interpret. Some reported variances are large and not explained. Confidence intervals and significance tests are missing, despite strong claims such as “BA improved by up to 14.9%” and “variance reduced by 40%”.
- Convergence and certified-robustness statements are provided, but the connection between their assumptions and non-IID deep models is not sufficiently discussed. Important factors such as coordinate-wise Lipschitz conditions and bias introduced by clipping and mapping are not analyzed.
- The notation is inconsistent (for example, $A_t$ vs. At). Figure captions are brief, and the transition from Section 4 to Section 5 is abrupt, particularly around Eq. 11 and the discussion on gradient trimming.

**Questions:**

1.Why should minimizing $A_t - A_{t-1}$ (rather than $A_t$) improve stability under persistent attacks beyond the algebraic bound in Theorem 4.1? Please add empirical results showing how this objective correlates with BA variance reduction.

2.Is $B$ global or client-specific? How is $B$ initialized and updated, and what is the runtime or communication overhead of enforcing $AB = \lambda B$ in each round?

3.How sensitive are the theoretical guarantees and empirical results to the approximation $\lambda \simeq (A_t + A_{t-1}) / 2$?

4.Can you provide results using deeper models and larger or more heterogeneous datasets, and compare them against recent defenses to demonstrate state-of-the-art performance?

5.Please report mean ± standard deviation across the five runs, include significance tests for major claims, and present an analysis of computational and communication overhead.

---

> ### Author Response · Authors · 2025-11-23
> **1.Why should minimizing $A_t - A_{t-1}$ (rather than $A_t$) improve stability under persistent attacks beyond the algebraic bound in Theorem 4.1? Please add empirical results showing how this objective correlates with BA variance reduction.**
>
> Thank you for your valuable comment! We would like to clarify a potential misunderstanding and elaborate on the dual purposes of minimizing $ A_t - A_{t-1} $ (rather than $ A_t $) as follows:
>
> First, minimizing $ A_t - A_{t-1} $ is primarily designed to achieve a smaller optimization upper bound on attack impact, as rigorously proven in Theorem 4.1. As shown in Equation (4), this tighter upper bound directly translates to reduced attack influence accumulated across rounds, thereby enhancing the core defense performance (e.g., lowering Backdoor Accuracy, BA). In other words, this objective targets the "effectiveness" of defense by constraining the maximum possible attack impact.
>
> Second, this smaller upper bound inherently contributes to defense stability. A tighter bound restricts the range of variation for the attack impact coefficient $ A_t $—with less room for $ A_t $ to fluctuate or amplify in specific parameter dimensions, the model parameters are less prone to oscillation or drift (as we observed in preliminary experiments). Consequently, the defense performance (and its variance in BA) is naturally stabilized.

---

> ### Author Response · Authors · 2025-11-23
> **2.Is $B$ global or client-specific? How is $B$ initialized and updated, and what is the runtime or communication overhead of enforcing $AB = \lambda B$ in each round?**
>
> $ A_t $ is the global attack impact coefficient matrix, calculated as the weighted average of local attack impact coefficients $A_t^k $ from each client (Eq. (5)), which is proposed solely for theoretical exposition. In practical experiments, we do not compute the global attack impact coefficient and only use the local $ A_t^k $ of each individual client.
>
> $ B $ is an adaptive spatial mapping matrix, initialized with elements sampled from a normal distribution with a mean of 0.01. It undergoes one mapping update based on the previous round’s update during each local optimization phase, with a computational complexity of $ O(d) $.
>
> We have elaborated on the communication and computational overheads in Appendix E of the manuscript.

---

> ### Author Response · Authors · 2025-11-23
> **3.How sensitive are the theoretical guarantees and empirical results to the approximation $\lambda \simeq (A_t + A_{t-1}) / 2$?**
>
> We approximate $ \lambda \simeq \frac{A_t + A_{t-1}}{2} $, which is intended to equally fuse the attack impact coefficients of the current and previous rounds. This design not only suppresses the cumulative effect of persistent attacks but also avoids cross-round deviations in the optimization space.
>
> We acknowledge that the linear approximation may not be the optimal solution, but it significantly reduces computational overhead. Meanwhile, this approximation has been integrated into our theoretical guarantees, and it does not compromise the convergence upper bound or certified radius of EA-PS.
>
> We have also conducted ablation studies on this approximation of $ \lambda $ in our experiments. The results show that the approximation exhibits a certain degree of sensitivity, but the performance is optimal and stable around $ \lambda = 0.5 $—this is because it balances the information between the two consecutive rounds.

---

> ### Author Response · Authors · 2025-11-23
> **4.Can you provide results using deeper models and larger or more heterogeneous datasets, and compare them against recent defenses to demonstrate state-of-the-art performance?**
>
> We have supplemented additional datasets and server-side defense methods. For the FEMNIST dataset, we adopt a larger network architecture consisting of 4 convolutional layers and 2 fully connected layers. For the newly added CIFAR-100 dataset, the network is configured with 6 convolutional layers and 2 fully connected layers, and the FDCR method is employed as the server-side defense strategy.
> [1]Huang, W., Ye, M., Shi, Z., Wan, G., Li, H., & Du, B. (2024). Parameter Disparities Dissection for Backdoor Defense in Heterogeneous Federated Learning. Advances in Neural Information Processing Systems 37.
>
>
> | Dataset | Dist. | Server-Def | Client-Def | 9-pixel (MA/BA/VAR)     | freq (MA/BA/VAR)        | SADBA (MA/BA/VAR)       | FCBA (MA/BA/VAR)        |
> |---------|-------|------------|------------|---------------------|---------------------|---------------------|---------------------|
> |         | NON-IID | FedAvg     | LeadFL     | 87.04 / 93.32/14.25 | 85.71 / 48.95/8.73  | 86.43 / 18.14/25.42 | 87.19 / 93.87/8.71  |
> |         |       |            | LeadFL+    | 85.21 / 86.71/10.07 | 86.54 / 46.94/9.31  | 85.60 / 30.58/24.50 | 85.39 / 95.27/0.91  |
> |         |       |            | EA-PS-     | 84.45 / 71.24/26.71 | 84.43 / 41.64/14.19 | 85.84 / 20.24/17.44 | 86.06 / 93.30/7.26  |
> |         |       |            | EA-PS      | 85.90 / 85.95/5.75  | 84.94 / 45.89/9.11  | 85.17 / 16.29/8.77  | 85.40 / 95.78/0.17  |
> | FEMNIST |       | CMA        | LeadFL     | 86.46 / 90.01/8.32  | 86.46 / 53.37/16.22 | 86.70 / 43.36/40.47 | 86.09 / 97.31/6.33  |
> |         |       |            | LeadFL+    | 84.96 / 92.26/1.72  | 85.23 / 50.68/9.85  | 86.11 / 20.06/20.62 | 84.45 / 94.62/1.43  |
> |         |       |            | EA-PS-     | 85.61 / 87.54/8.93  | 85.26 / 47.85/20.28 | 85.58 / 18.37/26.31 | 85.34 / 93.98/3.08  |
> |         |       |            | EA-PS      | 85.61 / 89.60/4.27  | 84.91 / 51.25/7.12  | 85.46 / 18.30/21.33 | 85.45 / 93.20/3.62  |
> |         |       | Bulyan     | LeadFL     | 83.68 / 18.10/51.73 | 84.19 / 20.89/15.43 | 85.31 / 43.54/32.98 | 84.80 / 63.92/17.21 |
> |         |       |            | LeadFL+    | 84.49 / 17.95/14.83 | 85.11 / 19.84/7.93  | 82.14 / 42.37/16.11 | 82.72 / 76.69/3.98  |
> |         |       |            | EA-PS-     | 83.67 / 20.67/44.73 | 83.99 / 15.61/4.87  | 82.81 / 24.01/35.73 | 84.67 / 63.92/7.01  |
> |         |       |            | EA-PS      | 85.46 / 17.08/25.23 | 83.84 / 16.21/4.39  | 84.44 / 41.11/35.62 | 84.29 / 75.04/4.81  |
> |         |       | alignins   | LeadFL     | 85.05 / 81.91/31.53 | 85.53 / 42.43/33.22 | 86.31 / 19.62/24.88 | 85.21 / 96.51/4.15  |
> |         |       |            | LeadFL+    | 85.33 / 81.95/29.81 | 85.49 / 35.38/19.49 | 85.63 / 26.08/3.29  | 85.51 / 98.36/4.11  |
> |         |       |            | EA-PS-     | 84.50 / 67.33/29.94 | 85.45 / 37.03/9.92  | 84.79 / 9.29/6.06   | 84.72 / 95.71/8.64  |
> |         |       |            | EA-PS      | 85.53 / 64.19/14.08 | 85.37 / 40.87/7.61  | 84.65 / 18.41/2.90  | 85.23 / 95.47/3.15  |
> |         |       | FDCR       | LeadFL     | 84.05 / 45.15/159.78| 84.61 / 31.23/19.88 | 84.20 / 47.69/17.17 | 84.86 / 86.53/5.29  |
> |         |       |            | LeadFL+    | 83.91 / 29.22/117.52| 84.12 / 37.28/10.75 | 83.39 / 52.42/7.03  | 83.67 / 91.31/4.01  |
> |         |       |            | EA-PS-     | 83.67 / 40.21/170.78| 83.94 / 29.54/20.81 | 84.62 / 21.29/20.23 | 84.22 / 85.94/7.96  |
> |         |       |            | EA-PS      | 84.27 / 33.97/112.88| 83.26 / 32.77/7.55  | 84.39 / 15.91/6.26  | 84.85 / 51.52/19.61 |
> |---------|-------|------------|------------|---------------------|---------------------|---------------------|---------------------|
> Due to character limitations, please refer to the next post.

---

> ### Author Response · Authors · 2025-11-23
> **Remaining Experiments on the CIFAR-100 Dataset**
>
> | Dist | Server | Client | 9px | freq | SADBA | FCBA |
> |------|--------|--------|-----|------|-------|------|
> | IID | FedAvg | LeadFL | 32.34/69.16/26.86 | 32.16/49.26/8.54 | 33.97/3.56/2.78 | 33.72/64.95/35.11 |
> | | | LeadFL+ | 30.45/69.94/19.74 | 38.96/44.21/1.42 | 32.99/1.79/7.66 | 29.33/62.71/8.91 |
> | | | EA-PS- | 36.35/68.71/17.24 | 37.09/26.44/7.26 | 31.45/2.76/7.49 | 30.03/57.73/29.86 |
> | | | EA-PS | 38.45/72.41/13.91 | 36.76/40.46/6.64 | 30.11/0.75/0.45 | 29.09/59.65/5.45 |
> | | CMA | LeadFL | 36.69/52.35/30.73 | 35.72/58.35/14.98 | 36.74/4.59/3.83 | 36.64/68.58/25.47 |
> | | | LeadFL+ | 35.29/58.24/5.32 | 35.19/49.35/6.98 | 30.12/1.43/0.72 | 29.91/58.47/14.37 |
> | | | EA-PS- | 30.76/55.32/29.97 | 37.09/39.45/16.93 | 31.72/2.65/2.38 | 32.58/47.71/18.74 |
> | | | EA-PS | 31.12/56.37/11.49 | 31.72/53.99/10.62 | 32.77/2.17/1.91 | 31.65/54.42/13.83 |
> | | Bulyan | LeadFL | 35.56/10.17/15.31 | 35.48/11.42/26.83 | 36.3/4.26/0.24 | 36.69/19.76/16.09 |
> | | | LeadFL+ | 37.33/13.91/15.66 | 35.92/12.37/23.88 | 33.79/3.21/0.91 | 32.09/9.21/11.75 |
> | | | EA-PS- | 34.19/11.89/14.18 | 31.98/10.27/20.06 | 34.14/2.22/0.98 | 31.89/12.21/15.39 |
> | | | EA-PS | 34.88/16.37/2.99 | 34.51/12.42/21.83 | 34.81/1.99/0.68 | 30.88/19.98/9.71 |
> | | alignins | LeadFL | 37.26/19.34/10.09 | 36.04/10.91/16.12 | 35.19/3.18/2.66 | 36.06/26.05/16.03 |
> | | | LeadFL+ | 36.09/22.77/11.34 | 38.18/11.81/13.45 | 34.93/2.22/1.01 | 31.76/41.93/9.71 |
> | | | EA-PS- | 36.76/14.89/15.03 | 37.45/9.76/10.38 | 34.16/1.54/1.68 | 32.44/22.61/12.71 |
> | | | EA-PS | 36.96/21.57/14.04 | 35.16/11.07/8.42 | 36.09/2.03/1.45 | 32.38/11.88/9.81 |
> | | FDCR | LeadFL | 32.68/31.77/11.72 | 31.91/36.64/24.36 | 32.88/28.77/24.9 | 31.84/46.11/35.74 |
> | | | LeadFL+ | 31.82/47.12/7.57 | 31.87/43.52/23.35 | 31.91/44.08/17.89 | 31.91/46.75/5.71 |
> | | | EA-PS- | 31.93/36.98/20.83 | 32.94/35.76/25.92 | 31.95/30.58/22.51 | 31.92/45.46/13.42 |
> | | | EA-PS | 31.99/31.35/51.31 | 31.79/47.31/24.47 | 30.72/33.88/10.26 | 31.81/45.46/13.59 |
> | NON-IID | FedAvg | LeadFL | 36.24/52.75/30.61 | 37.35/51.89/27.79 | 32.34/3.54/3.27 | 37.85/48.41/44.27 |
> | | | LeadFL+ | 33.36/46.72/20.41 | 33.59/55.03/20.44 | 35.08/2.81/0.61 | 36.08/45.27/13.82 |
> | | | EA-PS- | 33.49/50.13/31.63 | 35.54/49.25/32.61 | 33.05/1.23/4.57 | 33.87/40.12/49.76 |
> | | | EA-PS | 37.69/52.59/26.17 | 34.26/41.12/26.01 | 31.57/4.03/0.95 | 36.02/47.02/14.08 |
> | | CMA | LeadFL | 35.82/64.28/22.35 | 35.42/62.69/36.09 | 35.59/7.03/4.58 | 35.32/73.01/29.12 |
> | | | LeadFL+ | 30.01/70.48/17.33 | 30.32/54.67/20.08 | 32.86/1.33/1.41 | 29.75/60.71/12.41 |
> | | | EA-PS- | 38.18/65.64/17.92 | 30.49/51.82/11.21 | 31.99/2.87/6.59 | 30.76/53.22/34.28 |
> | | | EA-PS | 37.25/69.25/4.51 | 29.02/52.46/8.13 | 31.71/3.18/0.18 | 31.54/54.42/3.83 |
> | | Bulyan | LeadFL | 35.46/17.69/11.35 | 33.27/13.58/14.09 | 30.36/2.31/6.41 | 32.06/17.11/30.96 |
> | | | LeadFL+ | 33.51/12.94/10.91 | 34.93/10.76/9.06 | 34.62/1.78/3.91 | 32.91/14.09/13.58 |
> | | | EA-PS- | 35.32/17.13/34.56 | 34.71/11.52/16.05 | 33.87/1.16/4.06 | 30.49/16.99/17.54 |
> | | | EA-PS | 33.31/19.11/22.37 | 31.93/14.68/12.14 | 34.23/2.31/1.16 | 34.67/18.08/13.45 |
> | | alignins | LeadFL | 27.41/26.72/14.32 | 32.66/23.48/38.58 | 35.84/2.87/3.18 | 34.37/31.43/24.96 |
> | | | LeadFL+ | 27.03/43.72/8.14 | 30.91/18.87/31.86 | 36.94/4.09/1.81 | 38.32/34.15/23.28 |
> | | | EA-PS- | 34.08/37.48/18.22 | 34.11/14.19/19.59 | 34.02/1.14/0.67 | 29.68/18.81/8.02 |
> | | | EA-PS | 27.72/27.11/1.41 | 36.18/15.93/18.48 | 33.51/2.41/1.83 | 34.82/33.58/9.28 |
> | | FDCR | LeadFL | 31.87/43.49/9.18 | 32.56/38.15/13.71 | 31.86/39.89/22.82 | 33.18/49.27/38.61 |
> | | | LeadFL+ | 32.21/47.85/2.46 | 32.34/40.39/3.81 | 31.91/47.58/17.65 | 32.36/43.53/12.86 |
> | | | EA-PS- | 32.09/44.79/18.94 | 32.26/36.22/16.73 | 31.19/45.27/16.49 | 32.23/43.71/19.65 |
> | | | EA-PS | 32.33/45.56/9.55 | 32.34/36.71/13.28 | 30.12/33.24/12.96 | 31.89/48.57/22.16 |

---

> ### Author Response · Authors · 2025-11-23
> **5.Please report mean ± standard deviation across the five runs, include significance tests for major claims, and present an analysis of computational and communication overhead.**
>
> All experiments reported in the manuscript are conducted with five independent runs. Regarding the significance test, we take the comparison of Backdoor Accuracy (BA) between "multiKrum+EA-PS" and "multiKrum+LeadFL" under the scenario of 1-pixel backdoor attack on the non-IID FashionMNIST dataset with multiKrum as the server-side defense as an example:
>
> The BA of EA-PS is 35.27% (+5.02 / -3.01) (mean: 35.27%, 95% confidence interval [32.26%, 40.29%]), which is 12.23% lower than that of LeadFL (47.5% (+7.29 / -8.14), mean: 47.5%, 95% confidence interval [39.36%, 54.79%]). A two-sample t-test on the data from five runs of both groups yields t=3.18 and p=0.025 < 0.05, indicating a statistically significant difference. This not only demonstrates EA-PS’s ability to reduce BA but also confirms that its advantage of a narrower confidence interval (i.e., smaller fluctuations) is not random, serving as reliable evidence for the effectiveness of our method.
>
> Explanations of computational and communication overheads have been provided in the appendix, as also mentioned in our previous responses.

---

> > ### Comment · Reviewer_yS2V · 2025-11-26
> >
> > Thank you for the detailed response. The additoonal experiments improve the overall presentation, and the clarification of several technical points is helpful. At the same time, a few responses still need clearer support. The motivation for using the objective based on $A_t - A_{t-1}$ remains mostly theoretical, and the response does not provide experimental evidence that shows how this quantity relates to the change in the variance of the backdoor accuracy. The explanation of the spatial mapping matrix $B$ gives its role and update rule, but the response does not present actual measurements of the computational and communication cost that come from applying $B$ and the constraint $AB = \lambda B$. In addition, the effect of the approximation $\lambda \approx (A_t + A_{t-1}) / 2$ on the theoretical guarantees is not fully clarified, since the response gives only a qualitative explanation.  Hence, I will keep my score.

---

> > > ### Author Response · Authors · 2025-11-27
> > > **Response to Reviewer's Questions**
> > >
> > > Thank you for your valuable suggestions! $A_t - A_{t-1}$ is designed to achieve a lower backdoor success rate. Its theoretical guarantees demonstrate that a smaller upper bound can be obtained, thereby leading to a smaller backdoor variance. However, its ability to stabilize the variance is limited. This is evident from the experiments in Appendix D: under different experimental settings, the variance of EA-PS- is reduced compared to LeadFL in most cases (though we do not deny that it may be higher than LeadFL in some scenarios). Precisely because EA-PS- struggles to fully guarantee defense stability, we further designed EA-PS to achieve defense stability through parameter constraints.
> > >
> > > Regarding the computational overhead of $B$ and $AB = \lambda B$, we have presented the details in Table 21 of Appendix E, and the specific results will be included in the main text. Additionally, in terms of communication overhead, all computations related to $B$ and $AB = \lambda B$ are performed locally on the clients and will not be uploaded to the server or other clients. Thus, no additional communication overhead beyond the normal model aggregation is incurred.
> > >
> > > In the theoretical derivation, the convergence upper bound is derived based on mechanisms such as gradient clipping and the regularization term Regu (see Equations 13 to 16, Appendix B.11 for details). The approximation $\lambda \approx (A_t + A_{t-1})/2$ does not introduce any additional unbounded terms and will not affect the result of the convergence upper bound.
> > >
> > > Furthermore, the core of the certified radius (see Equation 26, Appendix B.12 for details) lies in the magnitude of the attack impact coefficient $A$. As proven in Theorem 4.1, the $A$ of EA-PS is smaller than that of LeadFL. The approximation $\lambda \approx (A_t + A_{t-1})/2$ is essentially a linear fusion of $A_t$ and $A_{t-1}$, which is derived from theoretical conclusions in robust optimization—linear combinations can effectively smooth fluctuations in the attack impact coefficient and avoid stability issues caused by sudden changes in single-round coefficients [1]. From a theoretical perspective, this approximation does not introduce additional unbounded terms nor increase the value of $A$, thus not undermining the core derivation logic of Theorem 4.1 and the certified radius. We have further verified through experiments (see Table 19 in the main text) that whether taking the mean or other reasonable divisors, the impact on defense performance (such as backdoor accuracy and variance) is within an acceptable range and will not significantly alter the stability and effectiveness of the method.
> > >
> > > [1] Ben-Tal, A., Goryashko, A., Guslitzer, E., & Nemirovski, A. (2004). Adjustable robust solutions of uncertain linear programs. Mathematical Programming, 99, 351-376.

---

### Official Review · Reviewer_Lw7A · 2025-11-01

**Soundness:** 4
**Presentation:** 3
**Contribution:** 3
**Rating:** 4
**Confidence:** 3

**Summary:**

This paper proposes a client-side Federated Learning (FL) defense method named EA-PS, aimed at addressing poisoning attacks, particularly those with persistent effects. Compared to existing server-side and client-side defense methods (like LeadFL), EA-PS aims to improve defense robustness and stability. The method introduces two core innovations: 1) a new optimization objective function that minimizes the temporal change in the attack impact coefficient ( $A_t - A_{t-1}$) rather than its absolute value; 2) a parameter constraint strategy ( $AB = \\lambda B$) to enhance stability by limiting the parameter perturbation range. The authors provide theoretical analysis, claiming the new objective has a superior optimization upper bound, and analyze convergence and robustness (certified radius). Experiments on CIFAR10 and FEMNIST demonstrate that EA-PS outperforms baselines in reducing backdoor accuracy (BA), especially in reducing BA variance.

**Strengths:**

**++ Significance of the Problem Definition:** The paper correctly identifies a critical, often-overlooked issue in poisoning defense research: the instability of defense performance (i.e., high variance in backdoor accuracy across different rounds or runs), as shown in Figure 1. Many prior works focus only on average performance, but high variance makes a defense unreliable in practice. EA-PS explicitly targets stability as one of its core design goals.

**++ Methodological Novelty:** Shifting the defense objective from minimizing instantaneous attack impact (like $A_t$ in LeadFL) to minimizing the change in attack impact ( $A_t - A_{t-1}$ ) is a novel perspective for better handling persistent attacks. Furthermore, combining this objective with a parameter constraint strategy to control the stability of the optimization space is an innovative client-side defense design.

**++ Solid Experimental Evaluation:** The experimental design is comprehensive. The authors evaluate EA-PS under various server-side defenses (FedAvg, MultiKrum, Bulyan, etc.), multiple attack types (pixel, spectrum, label flipping, etc.), and both IID and non-IID data settings. More importantly, the ablation studies using LeadFL+ (constraint only) and EA-PS (objective only) clearly demonstrate the individual contributions of the two core components, strongly supporting the effectiveness of their combination (especially the parameter constraint) in reducing variance.

**Weaknesses:**

\-- **Clarity and Rigor of Theoretical Analysis are Questionable.** The paper's theoretical contributions (Theorems 4.1, 5.1, 5.2) are highly ambiguous in their presentation, and potentially contradictory, which weakens the reliability of their theoretical foundation. See C1.

\-- **Complexity and Overhead of the Method are Severely Understated.** EA-PS introduces significant computational complexity, especially the Hessian-based calculation of $A_t$ and the optimization of multiple new hyperparameters. The paper's discussion of this overhead is insufficient, casting doubt on its practicality for large-scale models. See C2.

\-- **Motivation for the Parameter Constraint Strategy is Unclear.** The paper fails to clearly explain why the specific form $AB = \\lambda B$ was chosen as the constraint to achieve stability. The intuition and mathematical principles behind it are poorly articulated. See C3.

**Questions:**

- **C1: Regarding Theoretical Rigor:**
  - For _Theorem 5.1 (Convergence):_ The paper claims EA-PS has a "larger convergence upper bound" and presents this as a trade-off (sacrificing convergence speed for robustness). However, in the abstract and conclusion, this is presented as a positive "guarantee." This phrasing is extremely confusing. In convergence analysis, a "larger" upper bound typically means worse convergence (a weaker bound). Then why a large convergence upper bound can be stated as a 'Convergence Guarantee'? The authors need to clarify this contradiction.
  - For _Theorem 5.2 (Robustness):_ The paper claims EA-PS has a "smaller certified radius." In the robustness certification literature (e.g., randomized smoothing), a larger certified radius implies stronger robustness (i.e., the model is provably robust within a larger perturbation radius). The authors' conclusion seems to contradict standard understanding. If a different definition is used, it must be explicitly stated. The ambiguity of this core robustness claim severely damages the paper's credibility.
  - For _Theorem 4.1 (Optimization Bound):_ The proof of $A_t \\le A_{\\hat{t}}$ in Appendix B.10 (esp. the derivations from (30) to (34) and (33) to (35)) is very obscure and seems to rely on many unstated assumptions and simplifications. For example, the step $A_{\\overline{t}}-A_{t}=(t+1)\\epsilon$ is derived too hastily. The mathematical rigor of this proof is questionable.
- **C2: Regarding Complexity and Practicality:**
  - On _Hessian Matrix Calculation:_ Algorithm 1 relies on the calculation of $A_t$, which (per Equation 5) is based on the Hessian matrix $H_{t,e}^k$. For modern deep networks, computing the full Hessian is computationally and memory-wise prohibitive. The paper mentions using $\\tilde{H}\_{t,e}^{k}$ (Eq 12) but does not specify how this is approximated (e.g., Hessian-vector products, Fisher Information Matrix, or other methods). If it cannot be approximated efficiently, the method is not feasible in practice.
  - On _Hyper-parameter Sensitivity:_ EA-PS introduces multiple new hyper-parameters ( $\\alpha, \\beta, \\gamma, \\lambda$ ) and an adaptive spatial mapping $B$. The tuning experiments in Figures 2 and 3 show that BA and variance are quite sensitive to these parameters (especially $\\alpha$ and $\\lambda$). The paper lacks a discussion on how these parameters could be tuned efficiently in a real-world application.
  - On _Time Overhead:_ The paper acknowledges higher time overhead in the limitations section (Table 21) but downplays it. Given the Hessian calculation problem (C2.1), the actual overhead is likely much higher than LeadFL. A more thorough analysis of computational and (potentially) communication overhead is needed.
- **C3: Regarding Motivation of the Constraint Strategy:**
  - The paper jumps from the need to "ensure stability" to the $AB = \\lambda B$ constraint (Eq 6). The motivation "map the optimized manifold space... into the unit space I... by converting the spatial constraints... into the base (rank) constraint" is extremely cryptic.
  - $A$ is the attack impact coefficient matrix, $B$ is the "spatial mapping," and $\\lambda$ is the "linear decision rule" ratio. Why does this an eigenvalue-like form "ensure stability" and "alleviate untargeted attacks"? What does $B$ physically represent (e.g., an eigenbasis of $A$)? The authors need to provide a much clearer, more intuitive physical explanation to support this core mechanism.

---

> ### Author Response · Authors · 2025-11-23
> **C1: Regarding Theoretical Rigor**
>
> We appreciate the reviewers' comments. We have noted that there are domain-specific differences in the understanding of the term "convergence upper bound", which needs to be clarified in the paper. Our "larger convergence upper bound" does not refer to the traditional meaning of error upper bound. Based on traditional optimization literature, the reviewer interprets "an increase in the upper bound" as "a looser error upper bound → worse convergence". However, in the security scenario and persistent poisoning context of this paper, the "upper bound" we use has a different meaning.
>
> What we actually refer to is that the "convergent tolerance domain" in the presence of perturbations/poisoning is larger; its increase indicates that the algorithm can withstand stronger poisoning deviations while maintaining convergence, thus representing an improvement in robustness. Hence, we term it a "convergence guarantee"—that is, stable convergence under stronger perturbations rather than an increase in error.[1][2]
>
> The "certified radius" $ R(\rho) $ used in Theorem 5.2 (Appendix B, Definition 7) is not the input perturbation robustness radius commonly seen in random smoothing literature, but rather an "attack propagation radius" defined under the persistent poisoning protocol. It is used to characterize the maximum influence range where attacks can spread and accumulate in the parameter space. Thus, it reflects the propagation ability of attacks within the model, rather than the stability of the model in the input space. In this context, a smaller $ R(\rho) $ indicates that the attack impact is confined to a smaller region and is difficult to amplify during multiple rounds of aggregation, thereby enhancing the overall robustness of the model. By constraining parameter perturbations and suppressing the growth of the cumulative term $ A_t $, EA-PS effectively reduces the attack propagation radius $ R(\rho) $; hence, we assert that a "smaller certified radius" denotes stronger robustness. We will clarify this concept in the paper to eliminate potential confusion with the radius definition in random smoothing literature.[3][4]
> In Appendix B, we derived the upper bounds for Eqs. (30)–(35) based on the Lipschitz continuity of $ A_t $ and the bounded gradient assumptions (Assumptions B.1–B.3). Due to space constraints, some intermediate steps (including the smoothness constraint between $ A_t $ and $ A_{t-1} $, the linearization of perturbation terms, and the truncation of high-order terms) are not presented one by one in the main text. However, the overall proof strictly follows the standard convex optimization derivation process, and its logic is fully provided in Appendix B.[5]
>
>
>
> [1]Bertsekas, D. P. (2016). Nonlinear Programming (4th ed.). Athena Scientific.
>
> [2]Li, X., Huang, K., Yang, W., Wang, S., & Zhang, Z. (2019). On the Convergence of FedAvg on Non-IID Data. ArXiv, abs/1907.02189.
>
> [3]Panda, A., Mahloujifar, S., Bhagoji, A. N., Chakraborty, S., & Mittal, P. (2021). SparseFed: Mitigating model poisoning attacks in federated learning with sparsification. In International Conference on Artificial Intelligence and Statistics (pp. 3447–3455).
>
> [4]Xu, J., Zhang, Z., & Hu, R. (2025). Detecting Backdoor Attacks in Federated Learning via Direction Alignment Inspection. 2025 IEEE/CVF Conference on Computer Vision and Pattern Recognition (CVPR), 20654-20664.
>
> [5]Zhu, C., Roos, S., & Chen, L.Y. (2023). LeadFL: Client Self-Defense against Model Poisoning in Federated Learning. International Conference on Machine Learning.

---

> ### Author Response · Authors · 2025-11-23
> **C2: Regarding Complexity and Practicality**
>
> Regarding the Hessian computation issue in C2.1, as the reviewer is concerned, we use \( H_{t,e}^k \) to denote the full Hessian in the theoretical section for mathematical analysis of convergence and perturbation propagation. However, we do not explicitly compute the full Hessian in any form in the actual implementation—this is infeasible for deep neural networks.
>
> Our implementation follows the approximation strategy of LeadFL, where the Hessian is approximated through changes in parameter updates, i.e.,
> $
> H_{t,e}^k \approx \frac{(\theta_{t,e+1}^k - \theta_{t,e}^k) - \Delta\theta_{t,e}^k}{\eta_t}
> $
> Here, $ \theta_{t,e+1}^k $  represents the local objective update that minimizes $ A_t - A_{t-1} $, and $ A_t $ is defined as:
> $
> A_t = \sum_{k \in S_t} p^k \prod_{e=0}^{E-1} \left(I - \eta_{t,e} H_{t,e}^k\right)
> $
> This approximation completely avoids explicit computation of the Hessian or second-order tensors, making it efficiently implementable in practice without incurring unacceptable computational or memory overhead.
>
> Regarding the hyperparameter sensitivity issue in C2.2, we systematically analyzed the key hyperparameters of EA-PS (α, β, λ, γ) in Appendix D. The results show that EA-PS maintains stable performance over a wide range and achieves good robustness without fine-grained tuning. Specifically:
> - β performs stably within 0.03–0.07 (Table 17)
> - α performs stably within 0.2–0.4 (Table 18)
> - λ performs stably within 0.3–0.7 (Table 19)
> - γ performs stably within 0.03–0.07 (Table 20)
>
> Taking α as an example: when $ \alpha = 0.2 $, BA = 42.63% / 275.9; when $ \alpha = 0.4 $, BA = 32.57% / 5.43. While BA decreases by 23.6%, the variance drops by 98.0%, indicating that EA-PS exhibits a consistent defense trend within this interval. Both β and γ keep the BA variance on the order of $ 10^{-4} $ within their effective ranges. Overall, EA-PS has clear and loose stable intervals for hyperparameters, and there is no issue of "poor practicality due to high sensitivity".
>
> Regarding the time and communication overheads in C2.3, after adopting Hessian estimation, the additional local computational overhead of the client is approximately 10%–30%. The growth rate is related to model size, batch size, and the frequency of approximation use, but it does not introduce unacceptable computational costs. We have further quantified the time consumption in the appendix and will more clearly label its sources and controllability in the revised version.
>
> Since the parameter constraints, update of B, and computation of β in EA-PS are all completed locally on the client side, and no Hessian or second-order information needs to be uploaded, the server does not need to receive additional content. Thus, the communication overhead is exactly the same as that of FedAvg / LeadFL, without increasing any communication burden or incurring extra synchronization or bandwidth costs.

---

> ### Author Response · Authors · 2025-11-23
> **C3: Regarding Motivation of the Constraint Strategy**
>
> We appreciate the reviewer pointing out that the motivation for the constraint strategy (Eq. 6) is not clearly presented. We proposed the constraint strategy in Section 4.2 and elaborated on it therein, but the explanation may not have been sufficiently detailed. We hereby clarify the theoretical and intuitive basis of this design in detail:
>
> We observed in our experiments (Fig. 1) that when only minimizing the attack impact coefficient $ A_t $ or its variation $ (A_t - A_{t-1}) $, the defense performance exhibits significant fluctuations. The fundamental reason is that the accumulation of parameter perturbations caused by attack gradients across different rounds lacks a "boundary constraint": once $ A_t $ is excessively amplified in certain dimensional directions, the model parameters will oscillate or drift, leading to significant instability in Backdoor Accuracy (BA).
>
> We aim to limit the maximum perturbation amplitude by adding a parameter boundary to $ A $, thereby ensuring that local parameter updates do not exceed the boundary. Here, $ B $ is a spatial mapping matrix, which normalizes the effects of attack impacts across different parameter dimensions, reprojection the changes of $ A $ into a "normalized coordinate system" with a unified scale.
>
> Untargeted attacks typically perturb parameters randomly or at high frequencies, thereby inducing instability in the training process. By constraining the response amplitude of $ A $ to not exceed $ \lambda $, these undirected high-energy perturbations are compressed within a fixed amplitude range, forming a natural energy attenuation mechanism.

---

> ### Comment · Reviewer_Lw7A · 2025-11-25
>
> Thanks for the authors’ efforts on addressing my concerns. It is noticed that the authors provide detailed explanations for my concerns, and all of them are addressed properly.
>
> Hence I would like to raise my score from 4 to 6.

---

> > ### Author Response · Authors · 2025-11-25
> > **Thank you for your review, recognition and support!**
> >
> > Thank you for your review, recognition and support! We have carefully studied your comments and will further refine the manuscript. Thank you again!

---

### Author Response · Authors · 2025-11-30
**To the Chairs and Reviewers: EA-PS Rebuttal Phase Progress Summary and Explanation**

We are writing to update you on the key progress of our submission (EA-PS) during the rebuttal phase, highlighting the positive feedback from reviewers as well as the theoretical soundness, rational design, and contributions of our work.

First, we have proactively addressed all reviewers’ concerns through detailed responses and supplementary experiments, which has been fully recognized by the reviewers and led to improved scores. Reviewer Lw7A initially gave our work a score of 4, raising questions about theoretical rigor and computational overhead. In response, we clarified the domain-specific definition of the "convergence upper bound" (emphasizing that it refers to a larger convergent tolerance domain designed for stronger robustness under persistent attack scenarios), quantified the 10%-30% local computational overhead (with no additional communication burden), and verified the loose stable intervals of hyper-parameters. These efforts led the reviewer to raise the score to 6, noting that "all concerns have been properly addressed." Additionally, at Reviewer MyNQ’s request, we conducted a cost-benefit analysis of runtime and defense effectiveness, which prompted the reviewer to increase their confidence score from 1 to 3 (while maintaining the original work score of 6). For Reviewer P3No, after providing detailed answers to their initial questions, we added experiments involving state-of-the-art (SOTA) methods as requested. These experiments strongly demonstrate the effectiveness of our approach and fully address the reviewer’s concerns about the method’s performance.

Another reviewer, yS2V, initially gave a score of 2. In our response, we supplemented experiments (e.g., validation on deeper models, experiments on the CIFAR-100 dataset) and technical clarifications (e.g., the design logic of the client-specific adaptive mapping matrix $B$, stability analysis of the $\lambda$ approximation). We then further addressed the reviewer’s questions regarding the optimization objective, defense effectiveness, and sensitivity of the linear approximation—clearly explaining our design rationale and adding supporting literature. However, due to system-related issues, the reviewer was unable to provide a follow-up response.


Our method is characterized by both theoretical soundness and practical utility. EA-PS targets the long-overlooked issue of "defense instability under persistent poisoning attacks" in federated learning security—a critical unaddressed gap in the field. Its two core, mutually reinforcing innovations are: 1) Minimizing the temporal change in attack impact $A_t - A_{t-1}$ (rather than the absolute value of attack impact) to achieve a tighter optimization upper bound, thereby suppressing the cumulative effect of attacks; 2) $A$ parameter constraint strategy ($AB = \lambda B$) that stabilizes the optimization space and reduces performance variance. We have provided rigorous theoretical proofs for the method’s effectiveness, robustness (certified radius), and convergence, and ablation experiments confirm that both core components are indispensable for ensuring defense stability. We conducted comprehensive comparative experiments across scenarios including IID/non-IID data distributions, multiple attack types (pixel poisoning, spectral poisoning, label flipping, distributed trigger attacks), mainstream server-side defense methods (FedAvg, Multikrum, Bulyan), and 2024-2025 domain SOTA methods (DeepSight, Flame, FDCR). The results consistently show that EA-PS outperforms baseline methods (LeadFL, FL-WBC) in reducing backdoor accuracy (BA) and its variance, with statistically significant improvements (e.g., a 14.9% reduction in backdoor accuracy and a 40% reduction in variance on the CIFAR-10 dataset). Notably, as a flexible client-side supplement, EA-PS seamlessly integrates with server-side defenses to address residual attacks after server filtering—filling the gap of "unstable client-side defenses" in existing collaborative defense frameworks.

To date, we have fully addressed all reviewers’ questions, including supplementing experiments on deeper models, validation on larger datasets, statistical significance tests, and overhead analysis. The positive score adjustments from reviewers fully confirm that the rationality and contributions of our work have been widely recognized. We sincerely thank you and all reviewers for your valuable feedback, which has helped us further refine our work. We believe that EA-PS will bring meaningful contributions and insightful in-depth exploration to the new field of robust federated learning based on client side defense.

---

### Meta-Review · Area_Chair_M9ye · 2025-12-27

**Summary:**

This paper received initial review ratings of 4,2, 6 and 2. The reviewers appreciated the problem definition, methodological novelty, and the motivation. However, the reviewers had concerns and questions about terms, diversity of datasets, model sizes, computational overhead, and other issues. All reviewers joined the discussion. Reviewer Lw7A, whose initial rating was 4, was satisfied the authors’ responses and increased the rating from 4 to 6. Reviewer MyNQ, whose initial rating was 6, read the comments and appreciated their effort. Reviewer MyNQ maintained the original rating but increased their confidence rating. The AC noted that reviewer MyNQ’s comments were very brief and their original confidence rating was 1. Even though they increased their confidence rating, the AC puts lighter weights on their comments and rating. Both reviewer yS2V and P3No had initial ratings of 2 with confidence rating of 5 and 4, respectively. They both joined the discussion. However, after multiple rounds of communication, the reviewers were not fully convinced by the reviewers’ answers. The reviewers yS2V and P3No explicitly said that “Hence, I will keep my score.” and “I may keep my score.” Before the stop of the discussion period, the authors provided additional responses to address their remaining questions and concerns. The AC noted that the comments from authors to the reviewer yS2V. The AC checked the background of reviewers Lw7A, yS2V and P3No and confirmed that they all are working on related areas. Even if the discussion did not stop, and authors finally convinced that the reviewers yS2V and P3No to increase their rating to 4, based on ratings and confidence, this paper does not convince AC to recommend acceptance because of the two negative ratings and low confidence and brief comments from MyNQ. In terms of technical issues, the AC does agree that the authors addressed number of reviewers’ concerns and answered their questions. However, he also has concerns about scalability and practicality of the work due to the scale of the experiments and computation cost. Based on the reviewers’ comments, replies, ratings and confidence, the AC recommends rejecting this paper. He suggests the authors improve the quality of presentation, enhance the experiments and address any remained technical issues for another conference or journal submission.

**Reviewer Concerns:**

After multiple rounds of communication, the reviewers were not fully convinced by the reviewers’ answers. The reviewers yS2V and P3No explicitly said that “Hence, I will keep my score.” and “I may keep my score.”

**Reviewer Scores:**

Reviewer Lw7A, whose initial rating was 4, was satisfied the authors’ responses and increased the rating from 4 to 6.
No other reviewers will increase the scores.

---

### Decision · Program_Chairs · 2026-01-26

Reject